# No-Regret Algorithms in non-Truthful Auctions with Budget and ROI Constraints

## Abstract

Advertisers are increasingly using automated bidding to optimize their ad campaigns on online advertising platforms. Autobidding allows an advertiser to optimize her objective subject to various constraints. In this paper, we design online autobidding algorithms to optimize value subject to ROI and budget constraints.

We consider an item is being auctioned in each of $T$ rounds. We focus on one buyer with budget and ROI constraints in the stochastic setting: her value and highest competing bid faced are drawn i.i.d. from some unknown (joint) distribution in each round. We design low-regret bidding algorithms that bid on behalf of this buyer. Our main result is an algorithm with full information feedback (i.e., the highest competing bid is revealed after each round) that guarantees a near-optimal $\tilde{O}(\sqrt{T})$ regret with respect to the best Lipschitz function that maps values to bids. The class of Lipschitz bidding functions is rich enough to best respond to many correlation structures between value and highest competing bid, e.g., positive or negative correlation. Our result applies to a wide range of auctions, most notably any mixture of first- and second-price auctions. In addition, our result holds for both value-maximizing buyers and quasi-linear utility-maximizing buyers.

We also study the bandit setting, where the algorithm only observes whether the bidder wins the auction or not. In this setting, we show an $\Omega(T^{2/3})$ regret lower bound for first-price auctions, showing a significant disparity between the full information and bandit settings. We also design an algorithm with a regret bound of $\tilde{O}(T^{3/4})$ when the value distribution is known and is independent of the highest competing bid.

## Keywords

repeated auctions, online learning, first-price, budget constraint, ROI constraint

## 1 Introduction

With the growth of online advertising markets in terms of both complexity and scale, advertisers are increasingly turning towards autobidding to optimize their ad campaigns on online advertising platforms. Autobidding allows an advertiser to use an optimization algorithm to generate bids on her behalf, rather than manually bidding for each ad query. The advertiser provides high-level goals and constraints to the autobidder, which bids on her behalf in order to optimize her objective, while satisfying her constraints.

In this paper, we study the problem of designing algorithms for autobidding on behalf of a buyer. We consider a stochastic setting with $T$ rounds, in each of which one item is sold via an auction. In each round, the information of this round, including the buyer's value and the highest competing bid, are drawn i.i.d. from some unknown (joint) distribution. The autobidder submits a bid to the auction based on her value and the history. If the bid is at least the highest competing bid, the bidder wins the current

round and pays a price. The bidder has a budget constraint that limits the total payment, as well as a Return-on-Investment (ROI) constraint which requires that the total value in the winning rounds is at least a fraction of the total payment. These are the two most common constraints that bidders have in practice. In particular, ROI constraint captures similar constraints used in practice like target cost-per-acquisition (tCPA) and target return-on-ad-spend (tROAS)[1]. Our goal is to design online bidding algorithms that maximize the bidder's objective subject to both budget and ROI constraints. To quantify an algorithm's performance, we use (additive) regret against the objective value obtained by the best bidding strategy that knows the underlying distribution.

There has been a lot of recent work on the problem of designing algorithms for autobidding in stochastic settings. One line of work [6, 12] focuses on truthful auctions (e.g., second-price), which is proved to be much easier than non-truthful auctions due to technical reasons that we discuss later. A different line of work focuses on non-truthful auctions, with either a weak benchmark for regret, namely the best constant pacing (also sometimes called uniform bidding), where the bid is proportional to the value [14, 26], or have regret bounds that scale with the number of values and bids, which can be uncountably many [10].

In this paper, we study the problem of bidding in non-truthful auctions and design no-regret algorithms against a stronger benchmark than the best constant pacing – our algorithms have low regret compared to the *best Lipschitz bidding function that maps values to bids.* Due to the generality of Lipschitz functions this benchmark can best-respond to a range of different correlations between the buyer's value and the highest competing bid, e.g. positive correlation for some values and negative correlation for others.

**Our results and techniques.** We first consider the full-information setting where the bidder observes the highest-competing bid at the end of each round. We prove that there is an algorithm that can get near-optimal regret with respect to the best Lipschitz bidding function. The main result for this setting is as follows:

THEOREM 1.1 (INFORMAL VERSION OF THEOREM 4.1). *There is an algorithm that achieves $\tilde{O}(\sqrt{T})$[2] regret while satisfying both the budget and ROI constraints, with respect to the best Lipschitz bidding function. The result applies to various classes of auctions (see Assumption 3.1) including first-price auctions, second-price auctions and a hybrid of both. The result applies to both value and quasi-linear utility maximizing bidders.*

To the best of our knowledge, this is the first algorithm that achieves near-optimal regret bounds against the best Lipschitz bidding function for non-truthful auctions under budget and/or ROI constraints.[3] Our result applies to any input distribution under mild assumptions (see Section 2).

---

[1]For example, see the Google ads support page and Meta business help center
[2]$\tilde{O}(\sqrt{T}) = O(\sqrt{T} \cdot \text{poly}(\log(T)))$.
[3]See Section 1.1 for comparisons with prior works.

Our algorithm is based on the primal/dual framework for online learning with constraints [6, 7, 9, 10, 13, 20]. In this framework, to manage global constraints, the 'core' algorithm deploys two competing algorithms, each optimizing an unconstrained objective. On one hand, the primal algorithm picks an action (subsequently used in the actual bidding problem) to maximize a function similar to the Lagrangian of the problem. On the other hand, the dual algorithm picks Lagrangian multipliers to minimize the same function. Guarantees for this sequential unconstrained stochastic zero-sum game imply the guarantees for the original constrained problem.

While the dual algorithm uses a standard instance of Online Gradient Descent to pick the scalars that represent the Lagrangian multipliers, designing the primal algorithm is often much more complicated and requires knowledge specific to the original problem. We develop the primal algorithm for our main result in Section 3.

**Main Technical Challenges.** Below we list some of the main technical challenges that we need to tackle and give a brief outline of our approach to solving them.

*Lagrangian Maximization in Non-Truthful Auctions.* To better explain the challenge, we first consider the problem where the auction used is a second-price auction. The part of the Lagrangian function that depends on the primal algorithm's bid $b$ takes the following form (for either value or quasi-linear utility maximization): $r(b) = \mathbb{1}\left[b \geq d\right](\chi v - \psi d)$, where $v$ is the player's value, $d$ is the (unknown) highest competing bid, and $\chi, \psi$ are arbitrary non-negative numbers that depend on the Lagrange multipliers. Maximizing the above function turns out to be straightforward: using $b^* = v\frac{\chi}{\psi}$ implies[4] $r(b^*) = (\chi v - \psi d)^+$, which guarantees maximum reward. Since $b^*$ does not depend on the highest competing bid $d$, the primal algorithm can pick this bid to guarantee zero regret for maximizing the Lagrangian; this subsequently leads to low regret guarantees for the original problem with constraints.

In contrast to truthful auctions, for non-truthful auctions, the bid that maximizes the Lagrangian cannot be calculated without the highest competing bid. Therefore, the learner needs to learn the best function that maps values to bids. However, learning the best such function is unrealistic since it might be non-monotone and discontinuous. Instead, we focus on a class of functions with specific structures. Such a class used in previous work is the class of pacing multipliers, $\mathcal{F}_{\mathtt{mul}}$, that map values to bids by multiplying by a constant number. Instead, we focus on the more general class of Lipschitz continuous functions, $\mathcal{F}_{\mathtt{Lip}}$, which provide a much stronger benchmark to compete against, even in very simple settings where values and highest competing bids are independent. For example, if the highest competing bid is constant and the value is not, the best response is a fixed bid, which cannot be expressed by the class of pacing multipliers. In Appendix B we give a concrete example of this and include some additional discussion on the limitations of the pacing multiplier class $\mathcal{F}_{\mathtt{mul}}$.

The increased expressivity and complexity of $\mathcal{F}_{\mathtt{Lip}}$ over $\mathcal{F}_{\mathtt{mul}}$ can also be observed when considering finite approximations of them. $\mathcal{F}_{\mathtt{mul}}$ can be approximated with accuracy $\varepsilon$ using a set of size $\Theta(1/\varepsilon)$. If this approximation results in $O(T\varepsilon)$ error over $T$ rounds (this is not trivial, see our discussion on that next), along with many more simplifying assumptions, using standard online learning algorithms

---

[4]We denote $x^+ = \max\{0, x\}$.

we get $O(T\varepsilon + \sqrt{T\log(1/\varepsilon)})$ regret ($\sqrt{T\log K}$ is the regret of using $K$ different actions); optimizing over $\varepsilon$ we get $O(\sqrt{T})$ regret. In contrast, approximating $\mathcal{F}_{\mathtt{Lip}}$ with $\varepsilon$ accuracy requires a set of size $\exp(\Theta(1/\varepsilon))$, leading to $O(T\varepsilon + \sqrt{T\log(\exp(1/\varepsilon))})$ regret. This is $\tilde{O}(T^{2/3})$ if optimized over $\varepsilon$, which is suboptimal.

The near-optimal $\tilde{O}(\sqrt{T})$ regret is achieved by utilizing the structure implied by the finite subset of $\mathcal{F}_{\mathtt{Lip}}$, similar to [11, 18]. More specifically, we create a tree where the functions of the finite subset of $\mathcal{F}_{\mathtt{Lip}}$ are the leaves and smaller distance between two leaves implies more similarity between the two corresponding functions. This allows us to enhance the standard regret guarantees of learning algorithms to get the improved result, found in Section 3.3.

*Discretization Error and Safe bid.* Our algorithms are based on discretizing the bidding space of real numbers. However, two bids that are similar do not necessarily lead to similar reward, as the reward of a round is not a continuous function of the bid. This has been solved in previous works (e.g. Fikioris and Tardos [13], Han et al. [18]) for first-price by noticing that using bid $b + \varepsilon$ instead of $b$ still wins the auction and the price paid can only be $\varepsilon$ more. However, in this work we face one additional challenge: since our primal algorithms aim to maximize the Lagrangian, the reward of bid $b + \varepsilon$ might be negative, making bid $b$ much better if it *does not win* the auction. This means that the error of discretizing our action space is harder to handle. We tackle this by defining a general way of transforming bids to "safe bids" that guarantee non-negative reward that is at least as good as the original bid (Assumption 3.1), which is crucial to getting optimal regret rates.

*Time-Varying Range.* The Lagrangian function that the primal algorithm aims to maximize depends on the Lagrangian multipliers picked by the dual algorithm. Thus the primal algorithm's guarantees need to hold even against an adaptive adversary since no assumptions can be made for the dual algorithm's behavior, which adapts to the primal's decisions. While this challenge is not new to online learning algorithms, a new problem that we face is that the Lagrange multipliers control the range of the objective that the primal algorithm has to maximize. For technical reasons (which we discuss in Section 2), we cannot a priori upper bound these multipliers. This means that the primal algorithm needs to maximize a function whose range is time-varying and unknown. We develop algorithms that tackle this problem and offer regret bounds that match the bounds of algorithms that know this range in advance. We first solve this problem in Section 3.2 and use a technique that is very general and, we believe, is of independent interest.

*From Standard Regret to Interval Regret.* The 'core' algorithm requires that the primal and dual algorithms have low interval regret, i.e., low regret in every interval of rounds. This is not automatically achieved by classic algorithms, e.g., the Hedge algorithm [16] has linear interval regret. In Section 3.4, we offer a black-box reduction to reduce the problem of standard regret minimization to interval regret minimization with only $\tilde{O}(\sqrt{T})$ error, which also works for the above time-varying range problem.

**Bandit Information.** In Section 5 we consider the bandit information setting where the algorithm only observes whether the bid wins the auction or not and the price she pays if she wins. In sharp contrast to the full-information setting, we prove an $\Omega(T^{2/3})$ regret lower bound for first-price auctions even in a simple setting when

the value is constant. While this is known for quasi-linear utility maximization Balseiro et al. [5], no results are known for value maximization. Our lower bound is materialized in a very simple setting, as showcased in the theorem that follows.

THEOREM 1.2 (INFORMAL VERSION OF THEOREM 5.1). *No algorithm can always guarantee $o(T^{2/3})$ regret in value-maximizing first-price auctions with bandit information, even when the value is constant, the budget is $\Theta(T)$, and there is no ROI constraint.*

Our lower bound is based on a distribution of highest competing bids that has the following property: for (almost) every pair of values in the support, there is an optimal bidding strategy that uses only those values. A small adversarial modification in the distribution at a certain value ensures that (a) bidding at any other value is sub-optimal and (b) the bidder wastes many rounds on sub-optimal bids before finding the optimal one. This construction is inspired by the $\Omega(T^{2/3})$ lower bound of [23] for revenue maximization in posted-price mechanisms without constraints.

To complement our lower bound in Theorem 1.2, we present a $\tilde{O}(T^{3/4})$ regret upper bound in Theorem F.3.

**Tight satisfaction of the ROI constraint.** We remark that all our regret upper bounds satisfy the ROI constraint exactly but are based on similar results that approximately satisfy the ROI constraint (i.e. have sublinear violation with high probability). In Section 4, we present a black box reduction to turn any algorithm with approximate satisfaction into one with exact satisfaction.

Finally, we note that the focus of our $\tilde{O}(\sqrt{T})$ regret bounds in the full information setting (Section 3) is regret minimization. To get this optimal information theoretic bound our algorithms require exponential running time. In Appendix G, we present algorithms that require polynomial time to run and offer the same guarantees as Theorem 1.1 when the values and highest competing bids are independent across rounds.

## 1.1 Related work

The most relevant paper to ours is Castiglioni et al. [10]. The algorithm designed in our paper is based on the primal/dual framework in [10]; we briefly introduce the framework in Section 2. They also use the framework to design algorithms for bidding in first-price auctions with budget and ROI constraints, albeit only for a finite number of values and bids: their regret bound is $\tilde{O}(\sqrt{nmT})$ against the best bid per value, where $n$ is the number of values and $m$ is the number of bids. In addition, their algorithm satisfies the ROI constraint only approximately. In contrast, our results apply to continuous distributions and strictly satisfy the ROI constraint.

**Online bidding in non-truthful auctions.** Lucier et al. [26] design an algorithm for bidding in first and second price auctions under budget and ROI constraints, that implies welfare guarantees when used by every player (extending the result of Gaitonde et al. [17]). In addition, [26] prove that their algorithm, when used in a stochastic environment, has $\tilde{O}(T^{7/8})$ regret against the class of pacing multipliers while satisfying both constraints strictly. Fikioris and Tardos [14] also focus on welfare guarantees in first-price auctions when budgeted players use arbitrary algorithms that have no-regret against the class of pacing multipliers. In addition, they

design a full information algorithm that has $\tilde{O}(\sqrt{T})$ regret with respect to the same class in the stochastic environment. Finally, Wang et al. [31] study online learning in first-price auctions with budgets but focus only on the independent values and highest competing bids.

We defer further discussion about related work in Appendix A, were we discuss online learning in truthful auctions, online bidding without constraints, and online learning with or without budget constraints.

## 2 Preliminaries

We consider the setting where a single bidder participates in $T$ sequential auctions. In each round $t \in [T]$ there is a single item being sold; a pair $(v_t, d_t)$ is drawn i.i.d. from some unknown (joint) distribution $\mathcal{D}$, where $v_t \in [0, 1]$ indicates the bidder's value, and $d_t \in [0, 1]$ is the highest competing bid[5]. The bidder submits a bid $b_t$ based on her value $v_t$. The bidder wins this round if her bid is at least[6] the highest competing bid $d_t$; we denote $x_t = \mathbb{1}[b_t \geq d_t]$. If the bidder wins the auction, then she pays a price $p_t = p(b_t, d_t)$, where $p(b, d) \in [0, 1]$ is the payment function. For example, for first-price auctions, $p(b, d) = b$; for second-price auctions, $p(b, d) = d$; for any combination of the two auctions, $p(b, d) = q \cdot b + (1 - q) \cdot d$ for some $q \in [0, 1]$. We note that the payment function $p$ is fixed across all rounds and known to the bidder.

At the end of each round $t$, the bidder observes information about that round depending on the feedback model. In the full-information setting, the bidder observes the highest competing bid $d_t$ with which she can compute the outcome for any possible bid at this round. In the bandit-information setting, the bidder only observes whether she wins the auction or not (i.e. $x_t = \mathbb{1}[b_t \geq d_t]$) and the payment $p_t$ if she wins. Our results hold for different objectives of the bidder, who wants to maximize $\sum_{t \in [T]} u_t$, where $u_t$ is her per-round utility. The focus of our paper is value-maximizing, where $u_t = v_t x_t$, but our results also hold when the bidder has a quasi-linear utility, where $u_t = x_t(v_t - \nu p_t)$ for some $\nu \in [0, 1]$.

We assume that the bidder has a budget $B$. This is a strict upper bound on her total payment. Namely, it must hold that her total payment after $T$ rounds is at most $B$, i.e. $\sum_{t \in [T]} x_t p_t \leq B$. We define $\rho = \frac{B}{T}$ and note that w.l.o.g. we can assume that $\rho \leq 1$: any $\rho \geq 1$ implies that the bidder is effectively not budget constraint, since $p(\cdot, \cdot) \leq 1$. The bidder must also satisfy a Return-On-Investment (ROI) constraint: her total value in the winning rounds must be at least a fraction of her total payment: $\sum_t x_t v_t \geq \gamma \cdot \sum_t x_t p_t$, for some $\gamma \geq 1$. For the ROI constraint, we often allow approximate satisfaction where $\sum_t x_t(v_t - \gamma p_t) \geq -V$ and $V$ is the violation amount; often $V = O(\sqrt{T})$. W.l.o.g., we assume that $\gamma = 1$; any other $\gamma$ can be handled by rescaling the values[7].

**Benchmark.** We want low regret when competing against a class of bidding functions $\mathcal{F}$ that map values to bids. More specifically, we assume that for every $f \in \mathcal{F}$, $f$ maps $[0, 1]$ to $[0, 1]$ and we want the player's resulting utility to be close to her utility if she knew the distribution $\mathcal{D}$ in advance and she bid using the best fixed

---

[5]One can think of the highest competing bid as the highest bid among other bidders participating in the auction.

[6]Our results easily extend to other tie breaking rules.

[7]For any ROI $\gamma > 1$ we can rescale the values $v'_t := v_t/\gamma$.

functions from $\mathcal{F}$. Since the bidder has to satisfy certain constraints, the best response to a distribution $\mathcal{D}$ might be a distribution of functions over $\mathcal{F}$, not a single function. Our benchmark is the maximum expected average-per-round utility of the best distribution of functions from $\mathcal{F}$ that satisfies the constraints in expectation. For example, for a value maximizing player in first price auctions, i.e., $u_t = v_t \mathbb{1}[b_t \geq d_t]$ and $p_t = b_t$ we have

$$
\begin{aligned}
\mathsf{OPT} = \sup_{F \in \Delta(\mathcal{F})} \quad & \mathbb{E}_{f \sim F} \mathbb{E}_{(v,d) \sim \mathcal{D}} \left[ v \cdot \mathbb{1}[f(v) \geq d] \right] \\
\text{s.t.} \quad & \mathbb{E}_{f \sim F} \mathbb{E}_{(v,d) \sim \mathcal{D}} \left[ f(v) \cdot \mathbb{1}[f(v) \geq d] \right] \leq \rho \qquad (1) \\
& \mathbb{E}_{f \sim F} \mathbb{E}_{(v,d) \sim \mathcal{D}} \left[ (v - f(v)) \cdot \mathbb{1}[f(v) \geq d] \right] \geq 0
\end{aligned}
$$

For simplicity, we assume that the function $f(v) = 0$ always belongs in $\mathcal{F}$, making (1) always feasible. $T \cdot \mathsf{OPT}$ is an upper bound for the achievable total expected utility of any algorithm that satisfies the constraints [4, 9, 10]. We design algorithms that have low regret with respect to $T \cdot \mathsf{OPT}$ (where $\mathsf{OPT}$ is defined analogously depending on the buyer's objective and auction format).

**Primal/dual framework.** We now briefly describe the primal/dual framework where a constrained problem is solved by having two learning algorithms, the primal and the dual, compete against each other in a sequential unconstrained zero-sum game. Specifically we will look at the results of [10] who develop such a framework for budget and ROI constraints. We first discuss the assumptions required on the input distribution and how these relate to our setting of learning in sequential auctions. We then present the guarantees that the primal algorithm must satisfy to get guarantees for the original constrained problem. For simplicity, the rest of this section focuses on value maximization and presents all the assumptions in the context of auctions. We refer the reader to their paper for a more comprehensive description of their techniques.

First, we illustrate the need of some assumptions on the distribution $\mathcal{D}$ that generates $v_t, d_t$. Specifically, we assume that there exists a bidding function that on expectation leads to $\beta$ more value than payment, for some $\beta \geq 0$. Formally,

$$
\exists f \in \mathcal{F} : \quad \mathbb{E}_{(v,d) \sim \mathcal{D}} \left[ \left( v - p(f(v), d) \right) \mathbb{1}[f(v) \geq d] \right] \geq \beta. \qquad (2)
$$

Intuitively, this assumption implies that a learner who makes wrong decisions and violates her ROI constraint can recover this in later rounds. This assumption is similar to the one in [12], who examine value maximization in repeated truthful auctions. Their assumption is the same as (2) but for $f(v) = v$. This might seem stronger, since it implies (2) when the function $f(v) = v$ is contained in $\mathcal{F}$. However, the reverse also holds for truthful auctions, since $f(v) = v$ is optimal for maximizing quasi-linear utility.

We now show the basics of the primal/dual framework and the guarantees the primal needs to satisfy to get guarantees for the original problem. We first define the following function: $\mathcal{L}_t(b, \lambda, \mu) = \mathbb{1}[b \geq d_t](v_t - \lambda b + \mu v_t - \mu b) + \lambda \rho$. This function is inspired by the optimization problem in (1) (and would analogously be defined for other objectives/pricing functions). $\lambda, \mu$ are Lagrange multipliers that correspond to the budget and ROI constraint, respectively. The core algorithm of [10] runs two algorithms, the primal algorithm that picks bids and the dual that picks Lagrange multipliers. On every round $t$, the primal (resp. dual) algorithm picks $b_t$ (resp. $(\lambda_t, \mu_t)$)

aiming to maximize (resp. minimize) $\mathcal{L}_t(b_t, \lambda_t, \mu_t)$. Given their actions, each algorithm faces some regret. To get regret guarantees for the original problem (value maximization under constraints) the primal and dual algorithms must have with high probability low *interval regret*, i.e., have low regret over every interval $[\tau_1, \tau_2] \subseteq [T]$. Before formally defining this, we point out one subtle detail.

In general, regret bounds depend on the range of values that the objective function takes. This range for the primal algorithm in our setting can be as high as $2\mu_t + \lambda_t + 1$ in round $t$. This depends on the dual algorithm's actions $\lambda_t, \mu_t$, meaning we cannot know $\max_t\{2\mu_t + \lambda_t + 1\}$ in advance. One solution to this is to explicitly upper bound $\lambda_t$ and $\mu_t$. To get meaningful regret guarantees such an upper bound would be $\lambda_t, \mu_t \leq \frac{1}{\beta\rho}$. However, calculating $\beta$ requires $\mathcal{D}$, as seen in (2), which is unknown.

[9] show that the above issue can be circumvented, if the primal algorithm satisfies stronger *interval regret* bounds: with high probability, for every interval of rounds, the regret in those rounds is small with respect to the best fixed action in that interval and the maximum Lagrange multipliers seen so far. To formally define this, first define $M_t = \max_{t' \leq t}\{2\mu_{t'} + \lambda_{t'} + 1\}$. We require that the primal algorithm picks bids $b_1, \ldots, b_T$ such that for every $\delta \in (0, 1]$, with probability at least $1 - \delta$ it holds that for all $\forall [\tau_1, \tau_2] \subseteq [T]$ :

$$
\sup_{f \in \mathcal{F}} \sum_{t=\tau_1}^{\tau_2} \mathcal{L}_t(f(v_t), \lambda_t, \mu_t) - \sum_{t=\tau_1}^{\tau_2} \mathcal{L}_t(b_t, \lambda_t, \mu_t) \leq \mathsf{Reg}_\delta(T, M_{\tau_2}) \quad (3)
$$

Given the above guarantee for the primal and some mild conditions on Reg, we get the guarantees for the original problem. More specifically, we require that the dependence of $M$ in $\mathsf{Reg}_\delta(T, M)$ is not worse than quadratic. Using this condition, both the regret of the original problem and the violation of the ROI constraint are at most $\mathsf{Reg}_\delta(T, \delta, \frac{2}{\beta\rho})$. Formally, we have the following theorem.

THEOREM 2.1 (THEOREM 6.9 OF [10], ADAPTED TO AUCTIONS). *Assume (2) is true for $\beta > 0$, the dual algorithm is Online Gradient Descent, and (3) holds with $\mathsf{Reg}_\delta(T, M) \leq O(M^2 \mathsf{Reg}_\delta(T, 1))$. Then, for every $\delta > 0$, with probability at least $1 - 4\delta$, the regret of the core algorithm and the ROI violation is each at most $O\left( \mathsf{Reg}_\delta\left(T, \delta, \frac{2}{\beta\rho}\right) + \frac{1}{\beta\rho} \sqrt{T \log(T/\delta)} \right).$*

First, we note that [10] require a slightly more general condition than the one in (2) for $\beta > 0$; we present this in detail in Appendix C. Second, we note that if $|\mathcal{F}| = K$ was finite, we had bandit feedback, and a known upper bound $\bar{M}$ for $M_T$ then existing techniques would allow us to get $\mathsf{Reg}_\delta(T, M_t) = \tilde{O}(\bar{M}\sqrt{TK \log(1/\delta)})$. In the setting where there is no such upper bound, [10] offer an algorithm with $\mathsf{Reg}_\delta(T, M_t) = \tilde{O}(M_t^2 \sqrt{TK \log(1/\delta)})$. While this regret bound satisfies the requirements of Theorem 2.1, the quadratic dependence on $M_t$ is sub-optimal, making the resulting regret bound of the core algorithm to be proportional to $\frac{1}{\beta^2\rho^2}$. One of the contributions of the following sections is a general technique to get linear dependence of $M_t$ in $\mathsf{Reg}_\delta(T, M_t)$, similar to knowing $M_t$ in advance. This leads to much improved regret bounds that scale with $\frac{1}{\beta\rho}$, which is especially important when $\beta$ and $\rho$ are small.

# 3 Primal algorithm designs with full information

In this section, we design a primal algorithm that satisfies (3) for sequential auctions. Our goal is to pick bids to maximize the Lagrangian $\mathcal{L}_t(b_t, \lambda_t, \mu_t)$. For every round $t$, we define for simplicity $r_t(\cdot)$ to be the part of the Lagrangian that depends on the bid:

$$r_t(b) = \mathbb{1}\left[b \geq d_t\right]\left(\chi_t v_t - \psi_t p(b, d_t)\right) \tag{4}$$

where $d_t \in [0, 1]$ is the highest competing bid, $v_t \in [0, 1]$ is the value, $p(b, d_t)$ is the payment of bid $b$, and $\chi_t, \psi_t$ are non-negative numbers that depend on the Lagrange multipliers $\lambda_t$ (for budget constraint) and $\mu_t$ (for ROI constraint) of round $t$. For value maximizing, $\chi_t = 1 + \mu_t$ and $\psi_t = \lambda_t + \mu_t$. For quasi-linear utility, $\chi_t = 1 + \mu_t$ and $\psi_t = v + \lambda_t + \mu_t$ for some $v \in [0, 1]$. To present general results, we assume that $\chi_t, \psi_t$ are arbitrarily picked by an adaptive adversary with $\chi_t > 0$ and $\psi_t \geq 0$. We emphasize the lack of an upper bound on $\chi_t, \psi_t$ and recall (3), which requires that the regret scales with the largest $\chi_t, \psi_t$ seen so far. We overload the notation of $r_t(\cdot)$ to also take as an argument a bidding function $f : [0, 1] \to [0, 1]$, in which case $r_t(f) = r_t(f(v_t))$.

Our results apply to a wide range of price functions $p(\cdot, \cdot)$ that include any combination of first and second price. In particular, $p(\cdot, \cdot)$ needs to satisfy the following assumption which we explain after its formal statement.

*Assumption 3.1.* The pricing function $p(b, d)$ satisfies: (i) $p(0, 0) = 0$; (ii) $p(\cdot, d)$ is non-decreasing and 1-Lipschitz continuous[8] for all $d$; (iii) for every $t$, there exists a "safe" bid $b_t^\circ$ so that

(a) $r_t(b_t^\circ) \geq 0$ for all $d_t$.
(b) $b_t^\circ$ is a function of $v_t, \chi_t, \psi_t$ but not $d_t$.
(c) for every bid $b$ such that $\inf_{d_t \in [0,1]} r_t(b) < 0$, then for all $d_t$ it holds $r_t(b_t^\circ) \geq r_t(b)$.

While Conditions (i) and (ii) are standard assumptions on the payment function, Condition (iii) is less straightforward. In short, the safe bid guarantees that our algorithm will never use $b_t$ with $r_t(b_t) < 0$. For every $t$, we require that the function $r_t(b)$ has a "safe" bid $b_t^\circ$ such that: (a) $b_t^\circ$ guarantees a non-negative reward, (b) $b_t^\circ$ can be calculated using the information known before bidding, and (c) $b_t^\circ$ guarantees reward that is at least the reward of any other bid which has a negative reward for some $d_t$. We emphasize that (c) states that $r_t(b_t^\circ) \geq r_t(b)$ for all $d_t$ as long as $r_t(b) < 0$ for some $d_t$. In Appendix D.1, we show that any mixture of first and second price auction satisfies Assumption 3.1. In particular, the safe bid of Condition (iii) in this case is $b_t^\circ = \min\left\{\frac{\chi_t}{\psi_t} v_t, 1\right\}$.

## 3.1 Overview of the Primal Algorithm Design

We now state the main result of the section: There is an algorithm that has low interval regret with high probability. Our regret guarantee is with respect to the class of all $L$-Lipschitz continuous functions for any $L \geq 1$, which we denote with $\mathcal{F}$. In addition, our regret bound in rounds up to $t$ scale linearly with respect to the highest $\chi, \psi$ seen so far: $U_\tau = \max_{t \leq \tau}\{\chi_t, \psi_t\}$. Finally, we note that our algorithm takes into advantage the fact that $\chi_t, \psi_t$ are known before bidding in round $t$. This is important for various calculations,

---
[8] $g : \mathbb{R} \to \mathbb{R}$ is $L$-Lipschitz continuous if $|g(x) - g(y)| \leq L|x - y|$ for all $x, y \in \mathbb{R}$.

e.g., calculating the safe bid $b_t^\circ$ of a round. Outside of that, however, $\chi_t, \psi_t$ are picked adversarially and are not known before round $t$.

THEOREM 3.2. *Let $\mathcal{F}$ be the set of all $L$-Lipschitz continuous functions from $[0, 1]$ to $[0, 1]$ for some $L \geq 1$. Assume that the payment function satisfies Assumption 3.1. Assume that $\chi_t, \psi_t, d_t$ are picked by an adaptive adversary and $\chi_t, \psi_t$ are revealed after round $t - 1$. Let $U_{\tau_2} = \max_{t \leq \tau_2}\{\chi_t, \psi_t\}$. Then there exists an algorithm that generates bids $b_1, \ldots, b_T$ such that for all $\delta > 0$, with probability at least $1 - \delta$ it holds that for all intervals $[\tau_1, \tau_2] \subseteq [T]$*

$$\sup_{f \in \mathcal{F}} \sum_{t=\tau_1}^{\tau_2} r_t(f) - \sum_{t=\tau_1}^{\tau_2} r_t(b_t) \leq O\left(U_{\tau_2}\left(\sqrt{LT}\log T + \sqrt{T \log(T/\delta)}\right)\right).$$

Theorem 3.2 satisfies (3) as well as the conditions of Theorem 2.1 getting the following theorem.

THEOREM 3.3. *There is an algorithm for value or quasi-linear utility maximization when the payment function satisfies Assumption 3.1, such that for every $\delta > 0$, with probability at least $1 - \delta$ the algorithm has regret against the class of $L$-Lipschitz continuous functions and ROI violation at most $\frac{1}{\rho\beta} O(\sqrt{LT}\log T + \sqrt{T \log(1/\delta)})$.*

We note that the above algorithm, while providing an optimal (up to $O(\log T)$) information-theoretic bound, runs in exponential time. In Appendix G we present a polynomial time algorithm with matching regret which requires $v_t, d_t$ to be independent.

There are a couple of technical challenges in order to get low regret against $\mathcal{F}$ and prove Theorem 3.2. First, even for in-expectation regret bound, we cannot directly use a standard algorithm like Hedge, since $\mathcal{F}$ contains infinite actions. Instead, we work with finite approximations of $\mathcal{F}$; for accuracy $\varepsilon > 0$ let $\mathcal{F}_\varepsilon \subseteq \mathcal{F}$ such that

$$\forall f \in \mathcal{F}, \exists f_\varepsilon \in \mathcal{F}_\varepsilon : \quad f(v) \leq f_\varepsilon(v) \leq f(v) + \varepsilon, \ \forall v \in [0, 1] \tag{5}$$

i.e., for every $f$ in the original set $\mathcal{F}$ there exists some function $f_\varepsilon$ in the new set $\mathcal{F}_\varepsilon$ such that $f_\varepsilon$ is at least $f$, but is never greater by more than $\varepsilon$. Previous work shows that there is always an $\mathcal{F}_\varepsilon$ with $|\mathcal{F}_\varepsilon| \leq \exp(O(L/\varepsilon))$. [30, Corollary 2.7.2.]

The above bound on the cardinality of $\mathcal{F}_\varepsilon$ is exponential in $1/\varepsilon$. As explained in the introduction, using a standard online learning algorithm could only prove $O(T^{2/3})$ regret bounds. However, as Theorem 3.2 suggests, we can get much stronger $\tilde{O}(\sqrt{T})$ regret bounds. We solve this issue by utilizing the structure of $\mathcal{F}$, similar to [11, 18]. We create a hierarchical tree structure where the leaves of the tree are the functions of $\mathcal{F}$. Leaves whose distance is small represent bidding functions that are close in $L_\infty$ distance. Next, a non-leaf node above the leaf nodes can calculate a bidding function by combining the bidding functions of its children. Because its children are very 'close,' we can design algorithms so that the output has small regret with respect to its best child. Similarly, every non-leaf node does the same with its children, with nodes closer to the root having larger regret. This results in $O(\sqrt{T})$ regret instead of $O(T^{2/3})$. We develop this tree algorithm in Section 3.3.

In Section 3.2, we develop the algorithm that is used by the non-leaf nodes and that utilizes the proximity of its children's bids. We present this algorithm in the general language of online learning. The structure that this algorithm takes advantage of is that there is a "good" action: it is at most $\Delta$ sub-optimal compared to any other action, in every round. This leads to $O(\sqrt{\Delta T \log K})$ regret,

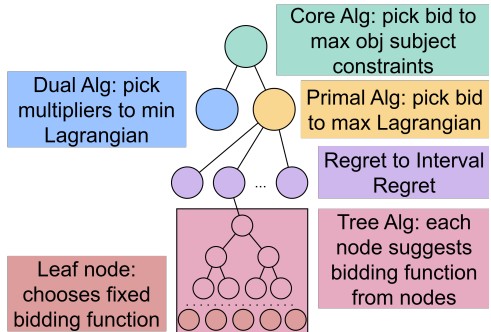

**Figure 1: The algorithm structure of the entire primal/dual framework in our setting.**

which offers a great improvement over the regret of $O(\sqrt{T \log K})$ that Hedge has when $\Delta \ll 1$.

Another technical challenge we face is regarding the discretization error. A naive assumption is that using $\mathcal{F}_\varepsilon$ instead of $\mathcal{F}$ leads to $O(\varepsilon)$ error every round. However, this is not the case: Let $f, f_\varepsilon$ as described in (5). Using $f_\varepsilon$ instead of $f$ in a round $t$ results in at most $\psi_t \varepsilon$ error if bidding $f(v_t)$ wins the auction (since $f_\varepsilon(v_t)$ also wins). However, if $f(v_t) < d_t \le f_\varepsilon(v_t)$ it might be the case that $r_t(f) = 0 \gg r_t(f_\varepsilon)$. For example, consider first-price with $\chi_t = \psi_t = 1$, $v_t = \varepsilon$, $f(v_t) = 1 - \varepsilon$, $f_\varepsilon(v_t) = 1$, and $d_t = 1 - \varepsilon/2$, in which case $r_t(f_\varepsilon) = -(1 - \varepsilon) \ll r_t(f) = 0$.

The safe bids of Assumption 3.1 solve this discretization issue. If a bid $b + \varepsilon$ is in danger of leading to a negative reward (i.e., $\inf_{d_t} r_t(b + \varepsilon) < 0$), we can use the safe bid instead to guarantee at least as good reward for any $d_t$, ensuring $O(\varepsilon)$ less reward than $r_t(b)$. We present this in Section 3.3. We note that one can circumvent this issue when maximizing quasi-linear utility with no constraints (i.e., when $\chi_t = \psi_t = 1$) by making sure that the class $\mathcal{F}$ contains only functions such that $f(v) \le v$, which is the solution of [18]. However, we cannot limit $\mathcal{F}$ in such a way since $\chi_t, \psi_t$ change dynamically.

Finally, to complete the proof of Theorem 3.2, in Section 3.4, we reduce the problem of bounding interval regret to bounding normal regret. The whole structure of our algorithm (including the primal/dual framework) is shown in Fig. 1.

Our resulting primal algorithm works by chaining multiple sub-algorithms, as shown in Fig. 1. The standard way to employ this chaining is to use each algorithm's outputting action. However, since our goal is a regret bound with high probability, the success of every algorithm is conditioned on the success of all its sub-algorithms. This creates noise in the high probability bounds, scaling with the number of sub-algorithms.

To overcome this challenge, we employ a different technique. Instead of an action, each algorithm outputs the distribution from which it would have sampled its action. These distributions satisfy a regret bound with probability 1, making chaining multiple algorithms much more stable since the sampling of an action happens only once and not for every sub-algorithm.

## 3.2 Time-varying Ranges and Good Actions

In this section we develop the algorithm that we need for the non-leaf nodes of the tree algorithm, which takes advantage of the proximity of the rewards of its actions. We present the algorithm

---

**ALGORITHM 1:** Hedge for time-varying ranges and good actions

**Input:** Total rounds $T$, actions $[K]$, sub-optimality of good action $\Delta$
Initialize cumulative reward of each action $R_0(a) = 0 \quad \forall a \in [K]$
**for** $t \in [T]$ **do**

    Receive range $[0, U_t]$ and calculate step size $\eta_t = \frac{1}{U_t}\sqrt{\frac{\log K}{T\Delta}}$
    Calculate probability distribution
    $p_t(a) = \exp\left(\eta_t R_{t-1}(a)\right) / \sum_{a'} \exp\left(\eta_t R_{t-1}(a')\right) \quad \forall a \in [K]$
    Sample and play action $a_t \sim p_t(a)$
    Receive rewards $r_t : [K] \to [0, U_t]$
    Update cumulative rewards $R_t(a) = r_t(a) + R_{t-1}(a) \quad \forall a \in [K]$

**end**

---

in the general language of online learning when there is a set $[K]$ of actions and an arbitrary reward function $r_t(\cdot)$ for every round $t$ which the learner observes after round $t$.

There has been extensive work on this setting, under various different assumptions that make the problem easier or harder. The contribution of this section, Theorem 3.5, is twofold. First, our algorithm is agnostic to the future range of the rewards. Specifically, we assume that the reward of round $t$ is in the range $[0, U_t]$ for some adversarially chosen $U_t > 0$ and the learner observes $U_t$ when picking her action $a_t$ in round $t$. We assume that $U_1 \le U_2 \le \ldots \le U_T = U$. If $U$ is known in advance, then using Hedge leads to regret that scales linearly with $U$. We achieve the same dependency without knowing $U$. This also improves the quadratic dependency on $U$ of previous work [10, Theorem 8.1].

Our second contribution in online learning when there is a $\Delta$-good action. The formal definition is in Definition 3.4, but simply put a $\Delta$-good action is at most $\Delta \cdot U_t$ sub-optimal compared to any other action in every round $t$. Definition 3.4 extends the original definition of [18] for time-varying reward ranges $U_1, \ldots, U_T$.

*Definition 3.4 (Good action).* For rewards $r_t : [K] \to [0, U_t]$, a $\Delta$-good action $g \in [K]$ satisfies $r_t(g) \ge r_t(a) - \Delta U_t, \forall a \in [K], t \in [T]$.

Taking $U_t = 1$ yields the definition of [18]. Note that $\Delta \in [0, 1]$ since $0 \le r_t(a) \le U_t$.

We now present the algorithm for the above setting. Our algorithm is the Hedge algorithm but with a carefully selected step size $\eta_t$. In particular, the probability of playing action $a$ in round $t$ is proportional to $\exp(\eta_t R_{t-1}(a))$, where $R_{t-1}(a) = \sum_{t' \le t-1} r_{t'}(a)$ and $\eta_t \propto 1/U_t$. We believe that this step size is of independent interest and can be used in any online learning setting to get regret bounds for time-varying ranges that match classical ones. Our regret bound is $O(U\sqrt{T\Delta \log K})$, which matches the one in [18], without assuming $U_t = U$ for every $t$ and that $U$ is known. The full algorithm can be found in Algorithm 1.

We now present our regret bound. As mentioned before, we bound the regret of the action distributions $p_1(\cdot), \ldots, p_T(\cdot)$ that Algorithm 1 generates instead of the regret of the sampled actions. This implies a matching bound on the expected regret. Using standard concentration inequalities we can get bounds with high probability. However, the most useful application of outputting distributions instead of actions is that chaining multiple algorithms becomes easier. This allows for stronger high-probability guarantees: the overall regret bound is dependent only on one sampling process, the one performed by the top-level algorithm.

**Theorem 3.5.** *Assume that an adaptive adversary picks the reward function $r_t : [K] \to [0, U_t]$ in every round $t$, where $U_1 \leq \ldots \leq U_T$. Assume that there is a $\Delta$-good action, with $\Delta \geq 4 \log K / T$. Then the action distributions $p_1, \ldots, p_T$ of Algorithm 1 guarantee $\forall \tau \in [T]$*

$$\max_{a \in [K]} \sum_{t \in [\tau]} r_t(a) - \sum_{t \in [\tau]} \sum_{a \in [K]} p_t(a) r_t(a) \leq 4 U_\tau \sqrt{T \Delta \log K}.$$

Algorithm 1 dynamically adapts to the time varying reward ranges due to $\eta_t \propto 1/U_t$. Because of this, if the adversary picks $U_t \gg U_{t-1}$, our algorithm adapts to the new ranges and picks the action of round $t$ accordingly. We believe this technique is very versatile and can be used to modify existing no-regret algorithms to work in the space of time-varying reward ranges. We show this in the bandit information setting in Theorem F.4. We defer the full proof of Theorem 3.5, along with the proofs of the other results of this section to Appendix D. In Appendix D we also include a high probability version of the above theorem, Theorem D.2 with $O(U \sqrt{T \Delta \log(T/\delta)})$ additional error. To retain regret that depends on $T\Delta$ instead of $T$, we cannot directly apply a standard concentration inequality and instead show that our rewards have low variance that depends on $\Delta$ to get the improved bound.

### 3.3 Algorithm for Lipschitz Bidding Functions

In this section we present the result that has low regret compared to the best $L$-Lipschitz function in the class $\mathcal{F}$, when maximizing the reward function $r_t(\cdot)$ as defined in (4).

The biggest novelty of this section is making sure that "similar" bidding functions result in similar reward. To that end, let $f, f_\varepsilon : [0, 1] \to [0, 1]$ such that $f(v) \leq f_\varepsilon(v) \leq f(v) + \varepsilon$ for all $v \in [0, 1]$. Bidding function $f$ is meant to represent an arbitrary function from $\mathcal{F}$ while $f_\varepsilon$ is meant to represent a function from $\mathcal{F}_\varepsilon$, the discrete cover of $\mathcal{F}$. We want to have low error when using $f_\varepsilon$ instead of $f$.

As we discussed in Section 3.1, $f_\varepsilon$ might lead to $r_t(f_\varepsilon) \ll r_t(f)$ if $f_\varepsilon(v_t) \geq d_t > f(v_t)$. We solve this issue by ensuring the bids we use never lead to negative reward, using Assumption 3.1. In our learning algorithm, each "action" (e.g. the $[K]$ actions in Theorem 3.5) represents a bidding function $f_\varepsilon \in \mathcal{F}_\varepsilon$. However, using the action that represents $f_\varepsilon$ will not lead to bidding $f_\varepsilon(v_t)$. Instead, when bid $f_\varepsilon(v_t)$ could lead to negative reward for some $d_t$ (recall $d_t$ is unknown when bidding) then we replace that bid with $b_t^\circ$, the safe bid of that round. Because of Assumption 3.1, we can both calculate when bid $f_\varepsilon(v_t)$ could lead to negative reward and guarantee that in that case the safe bid will lead to more reward. This trick guarantees that the reward of the action that corresponds to $f_\varepsilon$ is at most $\max\{\chi_t, \psi_t\} \cdot \varepsilon$ worse than $r_t(f)$. We emphasize that this step is necessary to guarantee bounded discretization error.

After using the above trick, the functions of $\mathcal{F}_\varepsilon$ are placed on a tree, with functions that are close in the tree implying that they are close in $L_\infty$ distance. The resulting algorithm is similar to the one in [18]. There are two key differences. First, the leaves of the tree suggest bids that use the above modified bids to ensure no-negative reward. Second, each non-leaf node combines the bids of its children to create a new bid distribution by using Theorem 3.5 to adapt to the time-varying ranges and take advantage of the $\Delta$-good arm. We next present the regret bound we get and defer the algorithm's full description and its proof in Appendix D.3.

**Theorem 3.6.** *For $\tau \in [T]$ let $U_\tau = \max_{t \leq \tau}\{\chi_t, \psi_t\}$ and $\mathcal{F}$ be the set of $L$-Lipschitz bidding functions for $L \geq 1$. There exists an algorithm that generates bid distributions $q_t$ that with probability 1,*

$$\sup_{f \in \mathcal{F}} \sum_{t=1}^{\tau} r_t(f) - \sum_{t=1}^{\tau} \sum_b q_t(b) r_t(b) \leq O\left(U_\tau \sqrt{LT} \log T\right).$$

### 3.4 Reduction from Regret to Interval Regret

We now show how to turn the regret bound of Theorem 3.6 to an interval regret bound. Recall that an interval regret bound is required to apply Theorem 2.1. We achieve the desired result by proving a general reduction from regret to interval regret, for any online learning problem with full information feedback.

Our reduction combines multiple online learning algorithms. Specifically, we consider $T$ different algorithms. The $t$-th algorithm ($t \in [T]$) "starts" in round $t$ and has low regret in the intervals $[t, t']$, for $t' > t$. For every interval, one of the $T$ algorithm has low regret but it is unclear how to get a *single* algorithm with low regret in *every* interval. We combine the output of each algorithm into a single distribution of action, using another online learning algorithm. For this meta-algorithm, we consider Algorithm 1, with the $K = T$ actions being the aforementioned $T$ algorithms. This results in low *interval regret* with a $O(\sqrt{T \log T})$ additional error.

Before mentioning our full result, we note one additional technique we have to use. Under the above description, it is unclear how the meta-algorithm handles inactive algorithms, i.e. algorithms that have not been started yet. This is partially resolved by constraining the meta-algorithm to sample only active algorithms. However, this does not decide what is the reward assigned to inactive algorithms (recall the final step of Algorithm 1). We resolve this by assigning them reward equal to the expected reward of the meta-algorithm under only the active algorithms. This reward structure ends up being equivalent to re-sampling an action if a inactive algorithm is sampled. We state the full result next.

**Theorem 3.7.** *Let $A$ be a set of actions and $r_t : A \to [0, U_t]$ be the reward function picked by an adaptive adversary. For every $\tau_1 \in [T]$ let $\mathcal{A}_{\tau_1}$ be an algorithm that generates distributions $\{q_t^{\tau_1}(\cdot)\}_{t \geq \tau_1}$ over the actions $A$ such that for all $\tau_2 \geq \tau_1$*

$$\sup_{a \in A} \sum_{t \in [\tau_1, \tau_2]} r_t(a) - \sum_{t \in [\tau_1, \tau_2]} \mathbb{E}_{a \sim q_t^{\tau_1}}[r_t(a)] \leq \mathsf{Reg}_{\tau_1}(\tau_2)$$

*Then there is an algorithm that can generate action distributions $q_1(\cdot), \ldots, q_T(\cdot)$ such that for all $[\tau_1, \tau_2] \subseteq [T]$:*

$$\sup_{a \in A} \sum_{t \in [\tau_1, \tau_2]} r_t(a) - \sum_{t \in [\tau_1, \tau_2]} \mathbb{E}_{a \sim q_t}[r_t(a)] \leq \mathsf{Reg}_{\tau_1}(\tau_2) + 4 U_{\tau_2} \sqrt{T \log T}.$$

The full description of our algorithm is in Algorithm 3, which is in Appendix D.4, along with the proof of Theorem 3.7. Combining the above theorem with Theorem 3.6 gives the following result.

**Corollary 3.8.** *In the same setting as Theorem 3.6, there is an algorithm that generates distributions over bids $q_1(\cdot), \ldots, q_T(\cdot)$ such that with probability 1, for every $1 \leq \tau_1 < \tau_2 \leq T$ it holds*

$$\sup_{f \in \mathcal{F}} \sum_{t=\tau_1}^{\tau_2} r_t(f) - \sum_{t=\tau_1}^{\tau_2} \sum_b q_t(b) r_t(b) \leq O\left(U_{\tau_2} \sqrt{LT} \log T\right)$$

Corollary 3.8 immediately implies Theorem 3.2 using standard concentration inequalities. We include these calculations in Appendix D for completeness.

## 4 Exact satisfaction of the ROI constraint

In this section we show how we can turn every algorithm that has an approximate satisfaction of the ROI constraint into one with exact satisfaction. This reduction (Lemma 4.2), together with Theorem 3.3, lead to the following theorem.

**THEOREM 4.1.** *In the same setting as Theorem 3.3, there exists an algorithm that always satisfies the budget and ROI constraints and with probability $1 - \delta$ has regret $\frac{1}{\beta^2 \rho} O(\sqrt{LT} \log T + \sqrt{T \log(1/\delta)})$.*

Theorem 4.1 follows from the following lemma, which combines two algorithms to get the desired reduction from approximate satisfaction of the ROI constraint to an exact one. One algorithm maximizes the objective and has low ROI violation with high probability. The other algorithm is much simpler: it maximizes value minus payment, i.e., the ROI constraint. By running the second algorithm for enough rounds, we can accumulate enough 'slack' to mitigate the violation caused by the first algorithm.

**LEMMA 4.2.** *Assume that there is an algorithm $\mathcal{A}_1$ such that for every $\delta > 0$, with probability at least $1 - \delta$, when running on any set of rounds $\mathcal{T}_1 \subseteq [T]$ it generates bids that have*

- *regret at most $\text{Reg}_\delta(|\mathcal{T}_1|)$.*
- *total ROI violation at most $V_\delta(|\mathcal{T}_1|)$.*

*and another algorithm $\mathcal{A}_2$ such that*

- *its bid $b_t$ in round $t$ satisfies $\mathbb{1}[b_t \geq d_t](v_t - p(b_t, d_t)) \geq 0$.*
- *for every $\delta > 0$, with probability at least $1 - \delta$, when run in any set of rounds $\mathcal{T}_2 \subseteq [T]$ it generates bids $\{b_t\}_{t \in \mathcal{T}_2}$ such that $\sum_{t \in \mathcal{T}_2} \mathbb{1}[b_t \geq d_t](v_t - p(b_t, d_t)) \geq Q_\delta(|\mathcal{T}_2|)$.*

*Consider bidding $0$ on rounds where the remaining budget is less than $1$, using $\mathcal{A}_1$ on rounds $t$ when $\sum_{t' \leq t-1} \mathbb{1}[b_t \geq d_t](v_t - p(b_t, d_t)) \geq 1$, and $\mathcal{A}_2$ on other rounds. This yields exact satisfaction of the ROI constraint. Moreover, for any $\delta > 0$, with probability at least $1 - 2\delta$ it has regret at most $\text{Reg}_\delta(T) + 2Q_\delta^{-1}(V_\delta(T) + 2)$ where $Q_\delta^{-1}(\cdot)$ is the inverse function of $Q_\delta(\cdot)$.*

The conditions of the two algorithms sketch the lemma's proof (the full proof is in Appendix E, along with the proof of Theorem 4.1). $\mathcal{A}_1$ violates the ROI constraint by at most $V(T)$ (for simplicity we drop the dependence on $\delta$) and the second algorithm, in order to make up that violation, needs to be run for about $Q^{-1}(V(T))$ rounds. This has two effects on the total regret. First, $\mathcal{A}_1$ is run for $Q^{-1}(V(T))$ fewer rounds, potentially missing out on any reward on those rounds. Second, $\mathcal{A}_2$ can use at most $Q^{-1}(V(T))$ of the budget, making the overall algorithm have to stop at most $Q^{-1}(V(T))$ rounds earlier. This entails the desired bound.

We briefly explain how we get Theorem 4.1 from Lemma 4.2. Theorem 3.3 satisfies the constraints for $\mathcal{A}_1$. $\mathcal{A}_2$ is the algorithm of Theorem 3.2 by setting $\chi_t = \psi_t = 1$ in $r_t(\cdot)$ of (4). This makes $\text{Reg}(T_1) = V(T_1) = \frac{1}{\rho\beta}\tilde{O}(T_1)$ and $Q(T_2) = \beta T_2 - \tilde{O}(\sqrt{T_2})$; the last equality follows from the assumption on $\beta$, (see Equation (2)).

**Necessity of $\beta$.** In Appendix E we present an example that shows dependency on $\beta$ is necessary in our regret bounds. Specifically, as $\beta$ decreases, the regret of any algorithm that exactly satisfies the ROI constraint increases polynomially in $1/\beta$. We show this for second-price auctions, to simplify the problem of regret minimization.

**THEOREM 4.3.** *In second-price auctions any algorithm with strict satisfaction of the ROI constraint cannot guarantee regret less than $(1 - 2\beta)\sqrt{T \frac{1}{2\pi\beta(1-\beta)}}$ when compared to the optimum LP value (e.g., see (1)) for every constant $\beta < 1/2$.*

## 5 Bandit information

In this section we study online learning in the bandit information setting. After bidding on round $t$, the bidder does not observe the highest competing bid $d_t$. Instead, she only gets to observe a Boolean value that indicates whether she wins this round, along with the price she paid if she wins, i.e., $\mathbb{1}[b_t \geq d_t]$ and $\mathbb{1}[b_t \geq d_t] p(b_t, d_t)$. Note that this setting is completely different between first-price and second-price auctions. In second-price, if the bidder wins a certain round, she gets to observe $d_t$ and the same is true whenever $p(b, d) = qb + (1 - q)d$ for $q > 0$. In contrast, when the bidder participates in strictly first-price auctions, she can never observe $d_t$. Next, we show that this is a crucial distinction that makes first-price auctions much harder: any algorithm has to suffer $\Omega(T^{2/3})$ regret, in contrast to the $\tilde{O}(\sqrt{T})$ regret bounds for second-price auctions [6, 12]. To complement the negative result, in Appendix F.2 we offer an algorithm with $\tilde{O}(T^{3/4})$ regret.

**Regret Lower Bound.** An $\Omega(T^{2/3})$ bound is known for quasi-linear utility maximization and no constraints by [5], which follows from the matching bound of [23]. Instead, we show a lower bound for value maximization with a budget constraint. In particular, our lower bound holds even if the value is fixed across rounds.

**THEOREM 5.1.** *There exists an instance in first-price auctions with no ROI constraint, $\rho = 1/4$, and $v_t = 1$ such that, any value-maximizing algorithm with only bandit feedback has regret $\Omega(T^{2/3})$.*

The problem described in Theorem 5.1 seems straightforward at first glance: the buyer wants to maximize the number of wins while adhering to the budget constraint. For example, if the CDF of $d_t$ is continuous and strictly increasing, one could consider that the bid $b^*$ such that $b^* \mathbb{P}[b^* \geq d_t] = \rho$, i.e., the bid that depletes the budget in expectation, is optimal. The problem of (approximately) finding such a bid $b^*$ is not hard, since it can be reduced to the problem of noisy binary search [21]. However, while this approach would work for second-price auctions, it does not work for first-price auctions. The reason is that our original hypothesis, that a single bid every round is optimal, is erroneous. There are strictly increasing and continuous CDFs for $d_t$ where, to maximize the number of wins while staying within budget, the bidder needs to use two distinct bids. In particular, the value of using the optimal two bids and the single optimal bid can be as big as a factor of 2.

In our proof we study a CDF $F$ for $d_t$ where there are infinite pairs of bids that, if mixed properly, attain the optimal solution. Next, an adversary perturbs $F$ by moving some mass to some bid $b^*$ from bid $b^* + \varepsilon$, making bid $b^*$ more 'valuable'. This entails that any algorithm that does not use $b^*$ enough faces considerable regret. The proof is completed by arguments similar to the ones in [23]: the bandit information that the bidder receives requires any algorithm to waste multiple rounds using sub-optimal bids before finding $b^*$. We include the full proof in Appendix F.1.

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

## A  Further Related Work

**Online bidding in truthful auctions**: Feng et al. [12] study online bidding in sequential truthful auctions under budget and ROI constraints in a stochastic environment. Their algorithm guarantees $\tilde{O}(\sqrt{T})$ regret with respect to the best bidding sequence and satisfies exactly both the ROI and budget constraints. Their results are an extension of the results of Balseiro and Gur [6], Balseiro et al. [7] where they study the same setting without ROI constraints. On the other hand, our paper studies a more general class of (possibly) non-truthful auctions and provides an algorithm that has the same regret guarantee. Balseiro and Gur [6], Balseiro et al. [7] also study the adversarial setting, where the value and the highest competing bid are not sampled by a stationary distribution but are picked by an adversary. In this setting, it is impossible to achieve regret that is sublinear in $T$, so they bound the competitive ratio, i.e. the multiplicative error, instead. Aside from truthful auctions, [7] also extend these guarantees to settings where the learner gets to observe every parameter of a round before picking a decision, e.g., in auctions observe the highest competing bid before bidding.

**Online bidding without constraints**: Another line of work studies online bidding without constraints. [18] study quasi-linear utility maximization in first-price auctions, while [11] study the maximization of arbitrary Lipschitz continuous functions. Both use an algorithm with a tree structure similar to ours; we comment on the similarity/differences in Section 3.3. Balseiro et al. [5] study contextual online learning, which result into a $\tilde{O}(T^{2/3})$ regret bound for quasi-linear utility maximizers in first-price auctions with bandit feedback. More recently, Kumar et al. [25] develop a $O(\sqrt{T})$ regret algorithm for first-price auctions with finite number of potential bids; they probive a $O(\log T)$ regret for stochasitc inputs. Kleinberg and Leighton [23] study online pricing, where the learner wants to learn how to price an item to maximize revenue; one of their results implies that the above regret bound is tight.

**Online learning with budgets**: *Bandits with Knapsacks* is a class of online learning problems where the learner has a general action space and multiple budget constraints [1, 2, 4, 13, 20, 22]. Bernasconi et al. [8], Kumar and Kleinberg [24], Slivkins et al. [29] study the same setting when the budget can also increase in some rounds.

**Online learning without constraints**: Finally, the problem of online learning without constraints has received extensive attention. [19] and [28] are excellent textbooks. The most commonly used algorithms for settings with finite number of actions are Hedge for full information feedback [16] and EXP3 for bandit feedback [3]. For online convex optimization (where there are infinite number of actions) the most commonly used algorithm is Online Gradient Descent [32].

## B  Limitation of Pacing multipliers

**Gap between Lipschitz bidding functions and Pacing multipliers.** Consider a value maximizer in first-price auctions with total budget $T/2$. Assume that her values are either $1/2$ or $1$, each with probability $1/2$. Also assume that the highest competing bid every round is $1/2$. The best Lipschitz bidding function is to bid $1/2$ every round: this stays within budget (in expectation) and leads to a total value of $\frac{3}{4}T$. In contrast, no fixed pacing multiplier can win all rounds and stay within budget. In particular the only pacing multipliers worth considering is bidding half or all of the value. In turns out that the best the bidder can do is bid her value with probability $1/2$ and half her value with probability $1/2$, leading to total value $\frac{5}{8}T$. This proves that the difference between the two benchmarks is $\frac{1}{8}T$, even in this very simple example.

**ROI Constraint for Pacing Multiplier in first-price auctions.** Another example that showcases the limitation of the best Pacing multiplier benchmark is value maximizing in first-price auctions. Here, an online learner who wants to compete against the best pacing multiplier does not have to consider the ROI constraint at all. This follows because the optimal such bidding would never use a pacing multiplier that would violate that constraint (i.e., bid above the value). In contrast, an optimal general bidding function for value maximization might bid above the bidder's value in some rounds, as long as it wins other rounds where the bid is sufficiently less than the value.

## C  Deferred Text of Section 2

We now present the assumption that [10] make for their results. They assume that there must exist a distribution $F \in \Delta(\mathcal{F})$ of functions such that if the player bids according to $F$ then for some $\alpha \geq 0$ on expectation: (a) the payment of the player is no more than $\rho - \alpha$, and (b) the value earned by the player is at least the payment plus $\alpha$. Formally, $\exists F \in \Delta(\mathcal{F})$:

$$
\begin{aligned}
&\mathop{\mathbb{E}}_{f \sim F} \mathop{\mathbb{E}}_{(v,d)} \left[ \left( v - p\big(f(v), d\big) \right) \mathbb{1}\left[ f(v) \geq d \right] \right] \geq \alpha \text{ and} \\
&\mathop{\mathbb{E}}_{f \sim F} \mathop{\mathbb{E}}_{(v,d)} \left[ p\big(f(v), d\big) \mathbb{1}\left[ f(v) \geq d \right] \right] \leq \rho - \alpha
\end{aligned}
\tag{6}
$$

The above conditions imply that substituting $F$ in problem (1), both constraints are satisfied with a slack of $\alpha$. In the absence of a ROI constraint, we would have $\alpha = \rho$. Given (6) and that $\alpha > 0$, the regret bounds of [10] depend on $\alpha$ and become worse as $\alpha$ becomes smaller.

We now prove how (2) implies (6) with $\alpha = \beta\rho$. We require that the function $f(v) = 0$ is included in $\mathcal{F}$ and bidding 0 guarantees 0 payment. Let $f_\beta$ be the bidding function that makes the guarantee in (2). We define $F$ to be the following distribution of functions: $f_\beta$ with probability $\rho$ and the 0 bid with probability $1 - \rho$. In this case we have

$$
\begin{aligned}
&\mathop{\mathbb{E}}_{f \sim F} \mathop{\mathbb{E}}_{(v,d)} \left[ \left( v - p\big(f(v), d\big) \right) \mathbb{1}\left[ f(v) \geq d \right] \right] \\
&\geq \rho \mathop{\mathbb{E}}_{(v,d)} \left[ \left( v - p\big(f_\beta(v), d\big) \right) \mathbb{1}\left[ f_\beta(v) \geq d \right] \right] \geq \rho\beta
\end{aligned}
$$

where the first inequality holds by $\mathbb{1}\left[ 0 \geq d \right] p(0, d) = 0$ and the second by (2). This proves the first inequality of (6). For the second

constraint of (6) we have

$$\mathbb{E}_{f \sim F} \mathbb{E}_{(v,d)} \left[ p(f(v), d)) \mathbb{1} \left[ f(v) \geq d \right] \right] ''$$

$$= \rho \mathbb{E}_{(v,d)} \left[ p(f_\beta(v), d)) \mathbb{1} \left[ f_\beta(v) \geq d \right] \right]$$

$$\leq \rho(1 - \beta)$$

where in the last inequality we used by (2) and $v \leq 1$. The above two inequalities prove (6) for $\alpha = \rho\beta$, as claimed.

## D  Deferred Proofs and Text of Section 3

In this section we present the deferred proofs and text of Section 3.

### D.1  Deferred Proof for safe bid in auctions

Here, we show that any mixture of first and second-price auction satisfies Assumption 3.1 and more specifically Condition (iii).

We start with first-price auctions, i.e., $p(b, d) = b$. For simplicity, we start with a simpler safe bid than the one we claimed in the main body of the paper, $b_t^\circ = 0$. Conditions (a) and (b) are obvious. For Condition (c) we notice that $\inf_{d_t} r_t(b) < 0 \iff \psi_t b > \chi_t v_t$. For such bids $b$, $r_t(b) \leq 0$ for all $d_t$ which makes Condition (c) follow from $r_t(0) \geq 0$.

We now move to second-price, i.e., $p(b, d) = d$, where a safe bid is $b_t^\circ = \min\left\{ \frac{\chi_t}{\psi_t} v_t, 1 \right\}$. Condition (b) follows from definition and Condition (a) follows by calculating $r_t(b_t^\circ) = (\chi_t v_t - \psi_t d_t)^+$. For condition (c), we notice that $\inf_{d_t} r_t(b) < 0 \iff b > \frac{\chi_t}{\psi_t} v_t$; this is because $r_t(b) < 0 \iff b \geq d_t > \frac{\chi_t}{\psi_t} v_t$ and the last inequality can be satisfied iff $b > \frac{\chi_t}{\psi_t} v_t$. For such $b$, $r_t(b_t^\circ) < r_t(b)$ would hold only if the bid $b_t^\circ$ could win an auction that $b$ would not, which is impossible since $b > b_t^\circ$; this implies (c).

We note that the above safe bid $b_t^\circ$ satisfies something much stronger than Condition (c), as we mentioned in Section 1. For every $d_t$, it holds that $b_t^\circ \in \arg\max_b r_t(b)$.

We now focus on the most general case, a mix of first and second price, i.e., $p(b, d) = qb + (1 - q)d$ for some $q \in [0, 1]$. Here the safe bid for second-price also works here: $b_t^\circ = \min\left\{ \frac{\chi_t}{\psi_t} v_t, 1 \right\}$. In this case $r_t(b_t^\circ) = (1 - q)(\chi_t v_t - \psi_t d_t)^+$, which proves Condition (b). Similar to before, $\inf_{d_t} r_t(b) < 0 \iff b > \frac{\chi_t}{\psi_t} v_t$ in which case an analysis like the ones above proves Condition (c).

### D.2  Deferred Proofs from Section 3.2

We first prove Theorem 3.5, the main theorem of Section 3.2.

PROOF OF THEOREM 3.5. We first shift the rewards: since Hedge remains the same if a (possibly time-dependent) constant is added to the rewards, we set for all $t, a$

$$r_t(a) \leftarrow r_t(a) + \Delta U_t - \max_{a' \in [K]} r_t(a')$$

This means that now the rewards are $r_t : [K] \rightarrow [-(1 - \Delta)U_t, \Delta U_t]$ and specifically for the good action, $r_t(g) \geq 0$.

Now let $W_t = \sum_a \exp(\eta_t R_{t-1}(a))$ and $\theta = \sqrt{\frac{\log L}{T\Delta}}$. Notice that the probability to play action $a$ in round $t$ is $p_t(a) = \frac{\exp(\eta_t R_{t-1}(a))}{W_t}$.

We have that

$$\frac{1}{W_t} \sum_a \exp(\eta_t R_t(a)) \tag{7}$$

$$= \sum_a \frac{\exp(\eta_t R_{t-1}(a))}{W_t} \exp(\eta_t r_t(a))$$

$$= \sum_a p_t(a) \exp(\eta_t r_t(a))$$

$$\leq \exp\left( \eta_t(1 - \eta_t U_t) \sum_a p_t(a) r_t(a) + \eta_t^2 U_t \Delta_t \right)$$

$$= \exp\left( \eta_t(1 - \theta) \sum_a p_t(a) r_t(a) + \theta^2 \Delta \right) \tag{8}$$

where we get the last equality by substituting $\eta_t = \frac{\theta}{U_t}$ and in order to prove the last inequality we first prove the following proposition.

PROPOSITION D.1. *Let $X$ be a random variable such that $\mathbb{E}[X] = x$, $c_1 \leq X \leq c_2$, with $c_2 \geq 0$. Then, for any $0 < \eta \leq \frac{1}{\max\{|c_1|, |c_2|\}}$,*

$$\mathbb{E}[\exp(\eta X)] \leq \exp\left( \eta x(1 - \eta(c_2 - c_1)) + \eta^2 c_2(c_2 - c_1) \right).$$

PROOF. Let $\sigma^2 = \mathbb{E}\left[ (X - x)^2 \right]$ and $c = \max\{|c_1|, |c_2|\}$. We have that

$$\mathbb{E}[\exp(\eta X)] = \mathbb{E}[\exp(\eta(X - x))] \exp(\eta x)$$

$$\leq \exp\left( \frac{\sigma^2}{c^2} \left( e^{\eta c} - 1 - \eta c \right) \right) \exp(\eta x)$$

$$\leq \exp\left( \frac{\sigma^2}{c^2} \eta^2 c^2 \right) \exp(\eta x)$$

$$= \exp\left( \sigma^2 \eta^2 \right) \exp(\eta x)$$

where the first inequality follows from Bernstein's inequality[9] and the last inequality follows by the fact that since $\eta c \leq 1$ we get $e^{\eta c} \leq 1 + \eta c + (\eta c)^2$. We now bound

$$\sigma^2 = \mathbb{E}\left[ (X - x)^2 \right]$$

$$\leq \mathbb{E}\left[ (c_2 - X)^2 \right] \leq (c_2 - c_1) \mathbb{E}[c_2 - X] = (c_2 - c_1)(c_2 - x)$$

where the first inequality follows from the fact that $\mathbb{E}\left[ (X - y)^2 \right]$ is minimized when $y = x = \mathbb{E}[X]$, i.e., when it is equal to the variance. The second inequality follows from $c_2 - X \geq 0$ and $X \geq c_1$. Rearranging proves the proposition. □

Now the inequality in (7) follows from the proposition by setting $X = r_t(a)$ with probability $p_t(a)$, $c_1 = -(1 - \Delta)U_t$, and $c_2 = \Delta U_t$ and noticing that $\eta_t = \frac{\theta}{U_t} \leq \frac{1}{U_t} = \frac{1}{c_2 - c_1} \leq \frac{1}{\max\{|c_1|, |c_2|\}}$.

Taking the logarithm of (7) we get

$$\frac{1}{\eta_t} \log\left( \frac{\sum_a \exp(\eta_t R_t(a))}{\sum_a \exp(\eta_t R_{t-1}(a))} \right) \leq (1 - \theta) \sum_a p_t(a) r_t(a) + U\theta\Delta$$

or equivalently

$$\frac{1}{\eta_t} \log \frac{\sum_a \exp(\eta_t R_t(a))}{K} - \frac{1}{\eta_t} \log \frac{\sum_a \exp(\eta_t R_{t-1}(a))}{K}$$

$$\leq (1 - \theta) \sum_a p_t(a) r_t(a) + U\theta\Delta \tag{9}$$

---

[9]See Lemma 7.26 in https://www.stat.cmu.edu/~larry/=sml/Concentration.pdf.

In the above equation we use the fact that

$$\frac{1}{\eta_t} \log \frac{\sum_a \exp(\eta_t R_t(a))}{K} \geq \frac{1}{\eta_{t+1}} \log \frac{\sum_a \exp(\eta_{t+1} R_t(a))}{K}$$

which follows from the fact that $\eta_{t+1} \geq \eta_t$ and the fact that the function

$$\left( \frac{1}{K} \sum_{i=1}^{K} x_i^{\eta} \right)^{1/\eta}$$

is increasing in $\eta$ for $x_i > 0$. This makes the previous inequality

$$\frac{1}{\eta_{t+1}} \log \frac{\sum_a \exp(\eta_{t+1} R_t(a))}{K} - \frac{1}{\eta_t} \log \frac{\sum_a \exp(\eta_t R_{t-1}(a))}{K}$$

$$\leq (1 - \theta) \sum_a p_t(a) r_t(a) + U\theta\Delta$$

Fix $\tau \in [T]$. We add the above for all $t \in [\tau - 1]$ along with (9) for $t = \tau$ and simplify the telescopic sum to get

$$\frac{1}{\eta_\tau} \log \frac{\sum_a \exp(\eta_\tau R_\tau(a))}{K} - \frac{1}{\eta_1} \log \frac{\sum_a \exp(0)}{K}$$

$$\leq (1 - \theta) \sum_{t \in [\tau]} \sum_a p_t(a) r_t(a) + U\theta T\Delta$$

Using the fact that $\sum_a \exp(\eta_\tau R_\tau(a)) \geq \exp(\eta_\tau R_\tau^*)$ (where $R_\tau^* = \max_a R_\tau(a)$) and substituting $\eta_\tau = \frac{\theta}{U_\tau}$ and $\theta = \sqrt{\frac{\log K}{T\Delta}}$ we get

$$R_\tau^* - U_\tau \sqrt{T\Delta \log K} \leq (1 - \theta) \sum_{t \in [\tau]} \sum_a p_t(a) r_t(a) + U_\tau \sqrt{T\Delta \log K} \tag{10}$$

Using the fact that $R_\tau^* \geq 0$ (since the reward of the good arm is always non-negative) we can use (10) to prove

$$\sum_{t \in [\tau]} \sum_a p_t(a) r_t(a) \geq -\frac{2}{1 - \theta} U_\tau \sqrt{T\Delta \log K} \geq -4U_\tau \sqrt{T\Delta \log K} \tag{11}$$

where we use the fact that $\theta = \sqrt{\frac{\log K}{T\Delta}} \leq 1/2$ since $\Delta \geq \frac{4\log K}{T}$. We rearrange the terms in (10) to get

$$R_\tau^* - \sum_{t \in [\tau]} \sum_a p_t(a) r_t(a) \leq 2U_\tau \sqrt{T\Delta \log K} - \theta \sum_{t \in [\tau]} \sum_a p_t(a) r_t(a)$$

$$\leq 2U_\tau \sqrt{T\Delta \log K} + 4\theta U_\tau \sqrt{T\Delta \log K}$$

$$\leq 4U_\tau \sqrt{T\Delta \log K}$$

where for the second inequality we used (11) and for the final inequality we used $\theta \leq \frac{1}{2}$. This proves the theorem. □

We now state and prove the high probability version of Theorem 3.5.

THEOREM D.2. *In the same setting as Theorem 3.5, Algorithm 1 guarantees the following high probability bound: for every $\delta > 0$ probability at least $1 - \delta$, it holds that*

$$\forall \tau \in [T] : \max_{a \in [K]} \sum_{t \in [\tau]} r_t(a) - \sum_{t \in [\tau]} r_t(a_t)$$

$$\leq 4U_\tau \left( \sqrt{T\Delta \log K} + \max \left\{ \sqrt{T\Delta \log(T/\delta)}, \log(T/\delta) \right\} \right)$$

The high probability bound does not follow from a simple application of the Azuma-Heoffding inequality, since the range of $\sum_t r_t(a_t)$ can be $\Omega(UT)$ making the resulting error $O(U\sqrt{T\log(1/\delta)})$ and not $O(\sqrt{T\Delta \log(1/\delta)})$ like in the above. Instead, we use Freedman's inequality, which offers a bound based on $\sum_t \text{Var}[r_t(a_t)]$ which we

prove is $O(U^2 T\Delta)$. This allows us to get the improved dependence on $\Delta$.

PROOF OF THEOREM D.2. For every $t$, let $X_t = \sum_a p_t(a) r_t(a)$ and $Y_t = \sum_{\tau=1}^{t} (X_\tau - r_\tau(a_\tau))$. The theorem follows by showing that for every $\delta > 0$

$$\mathbb{P}\left[ \forall \tau \in [T] : Y_\tau \leq 4U_\tau \max\left\{ \sqrt{T\Delta \log(T/\delta)}, \log(T/\delta) \right\} \right] \geq 1 - \delta.$$

We are going to use Freedman's inequality [15, Theorem 1.6] on the sequence $Y_0, Y_1, \ldots$ which we first prove is a martingale with respect to the the history of the rounds (we denote with $\mathbb{E}[t]\cdot$ the expectation conditioned on the history of the rounds up to $t$, i.e., the actions that the player and the adversary has take up to $t$): for every $t \geq 1$

$$\mathbb{E}_{t-1}[Y_t - Y_{t-1}] = \mathbb{E}_{t-1}[X_t - r_t(a_t)] = 0$$

where the last inequality holds because $a_t = a$ with probability $p_t(a)$. This proves the martingale property. We now notice that $|Y_t - Y_{t-1}| \leq U_t$ and that

$$\mathbb{E}_{t-1}\left[ (Y_t - Y_{t-1})^2 \right]$$

$$= \mathbb{E}_{t-1}\left[ (X_t - r_t(a_t))^2 \right]$$

$$\leq \mathbb{E}_{t-1}\left[ (\Delta U_t - r_t(a_t))^2 \right]$$

$$\leq U_t \mathbb{E}_{t-1}[\Delta U_t - r_t(a_t)] = \Delta U_t^2 - U_t X_t$$

where we notice that in the first inequality we use the fact that $\mathbb{E}_{t-1}[r_t(a_t)] = X_t$ and that $\mathbb{E}_{t-1}\left[ (X_t - r_t(a_t))^2 \right]$ is the conditional variance of $r_t(a_t)$, which means that $\mathbb{E}_{t-1}\left[ (c - r_t(a_t))^2 \right]$ is minimized when $c = \mathbb{E}_{t-1}[r_t(a_t)] = X_t$. For the second inequality we used that $-(1 - \Delta)U_t \leq r_t(a) \leq \Delta U_t$. We now have that

$$\sum_{t \in [\tau]} \mathbb{E}_{t-1}\left[ (Y_t - Y_{t-1})^2 \right]$$

$$\leq \sum_{t \in [\tau]} (\Delta U_t^2 - U_t X_t) = U_\tau^2 T\Delta - U_\tau \sum_{t \in [\tau]} X_t$$

$$\leq U_\tau^2 T\Delta - U_\tau \left( \max_{a \in [K]} \sum_{t=1}^{T} r_t(a) - 4U_\tau \sqrt{T\Delta \log K} \right) \quad \text{(Theorem 3.5)}$$

$$\leq U_\tau^2 T\Delta + 4U_\tau^2 \sqrt{T\Delta \log K} \quad (r_t(g) \geq 0)$$

$$\leq 3U_\tau^2 T\Delta \quad \left( \log K \leq \frac{T\Delta}{4} \right)$$

Now using Freedman's inequality gives us that for all $\varepsilon > 0$

$$\mathbb{P}[Y_\tau < \varepsilon] \geq 1 - \exp\left( -\frac{\varepsilon^2/2}{3U_\tau^2 T\Delta + U_\tau \varepsilon/3} \right)$$

Let $\delta > 0$ such that

$$\varepsilon = U_\tau \max\left\{ \sqrt{12}\sqrt{T\Delta \log(1/\delta)}, \frac{4}{3}\log(1/\delta) \right\}. \tag{12}$$

This and a union bound over all $\tau$ proves the lemma as long as we prove that

$$\frac{\varepsilon^2/2}{3U_\tau^2 T\Delta + U_\tau \varepsilon/3} \geq \log(1/\delta)$$

or equivalently

$$\varepsilon^2 \geq 6U_\tau^2 T\Delta \log(1/\delta) + \frac{2}{3}U_\tau \varepsilon \log(1/\delta)$$

The above inequality is true because, by the definition of $\delta$ in (12),

$$\varepsilon^2 \geq 12 U_\tau^2 T \Delta \log(1/\delta) \quad \text{and} \quad \varepsilon \geq \frac{4}{3} U_\tau \log(1/\delta)$$

Multiplying the second inequality with $\varepsilon$ and adding them gives us the desired bound on $\varepsilon^2$. $\qquad\square$

## D.3 Deferred Proof and Algorithm of Section 3.3

As we mention at the beginning of Section 3, we focus on subsets of $\mathcal{F}$ that approximate $\mathcal{F}$ and have finite cardinality. We use a slightly different notation and define the sets $\mathcal{F}_0, \mathcal{F}_1, \ldots, \mathcal{F}_M$ (for some $M \in \mathbb{N}$) such that for every $i = 0, \ldots, M$ it holds that

$$\forall f \in \mathcal{F}, \exists f' \in \mathcal{F}_i : \quad f(v) \leq f'(v) \leq f(v) + 2^{-i}, \ \forall v \in [0, 1]. \tag{13}$$

As we mentioned in Section 3.3, instead of directly using a function $f_M \in \mathcal{F}_M$ and its bid $f_M(v_t)$ in round $t$, we modify that bid, since for certain values of $d_t$, $r_t(f_M)$ might be negative. This is where we use Assumption 3.1 and define

$$b_t^{f_M} = \begin{cases} f_M(v_t), & \text{if } \inf_{d_t} r_t(f_M(v_t)) \geq 0 \\ b_t^\circ, & \text{if } \inf_{d_t} r_t(f_M(v_t)) < 0 \end{cases} \tag{14}$$

where $b_t^\circ$ is as defined in Assumption 3.1, depending on the payment function. $b_t^{f_M}$ is never worse than $f_M(v_t)$, since $r_t(b_t^{f_M}) \geq r_t(f_M)$ and $r_t(b_t^{f_M}) \geq 0$, which follow from Assumption 3.1. This guarantees that our reward is always non-negative, which as we discussed in Section 3.1 is the key to bidding according to $\mathcal{F}_M$ and have $O(2^{-M})$ error compared to using the full $\mathcal{F}$.

We use the functions in $\mathcal{F}_0, \mathcal{F}_1, \ldots, \mathcal{F}_M$ to create a tree. For $i = 0, \ldots, M$ the $i$-th level of the tree has $|\mathcal{F}_i|$ nodes, each representing an $f_i \in \mathcal{F}_i$. We connect these nodes by defining each node's parent: for every $i = 1, \ldots, M$ and $f_i \in \mathcal{F}_i$, $P(f_i)$ is the parent of $f_i$ such that $P(f_i) \in \mathcal{F}_{i-1}$ and $\|f_i - P(f_{i-1})\|_\infty \leq 2^{-i+1}$. We note that such a $P(f_i)$ always exists because of (13). The parent function $P(\cdot)$ subsequently defines the children $C(f_i)$ and leaves $\mathcal{L}(f_i)$ of a node $f_i$, $i = 0, \ldots, M - 1$. Finally, we assume that $|\mathcal{F}_0| = 1$ (which does not violate (13)), making the tree have a unique root.

Using this tree, in every round $t$ the algorithm creates a distribution of bids in the following way. First, each leaf $f_M \in \mathcal{F}_M$ calculates the bid $b_t^{f_M}$ as defined in Eq. (14); this bid defines a trivial bid distribution $q_t^{f_M}(\cdot)$ which suggests the bid $b_t^{f_M}$ with probability 1. On the $i$-th level where $i < M$, each non-leaf node $f_i \in \mathcal{F}_i$ uses a distribution $p_t^{f_i}(\cdot)$ over its children $C(f_i)$ and the bid distribution $q_t^{f_{i+1}}(\cdot)$ of each child $f_{i+1} \in C(f_i)$ to define its own bid distribution $q_t^{f_i}(\cdot)$. This node $f_i$ runs an instance of Algorithm 1 to calculate the distribution over its children $p_t^{f_i}(\cdot)$. This recursive calculation defines the bid distribution of the root, $q_t^{f_0}(\cdot) = q_t(\cdot)$, which is the output of the algorithm.

We show the full algorithm in Algorithm 2, which has one slight modification compared to the process described above: each non-leaf node considers one more bid in addition to the bids suggested by its children. This bid is the maximum bid suggested by any of its leaves. This additional bid is key to the guarantee of our algorithm since it is the action that is $\Delta$-good (recall Definition 3.4). This allows us to use the improved regret bound of Theorem 3.5 which

is crucial: a node on the $i$-th level can have as many as $\Theta(\exp(2^i))$ children, which, if $\Delta$ was a constant, would lead to terrible regret bounds. Instead, we show that the additional action makes $\Delta \approx 2^{-i}$ which makes the regret of every non-leaf node $\tilde{O}(\sqrt{T})$.

---

**ALGORITHM 2:** Tree algorithm

**Input:** Number of rounds $T$, Lipschitz parameter $L$, number of levels of tree $M$

For every $i = 0, 1, \ldots, M$, let $\mathcal{F}_i$ be as described in (13)

```
// Description of tree structure
```
Let $P(f_i) \in \mathcal{F}_{i-1}$ such that $\|P(f_i) - f_i\|_\infty \leq 2^{-i+1}$, $\forall i = 1, \ldots, M$,
$f_i \in \mathcal{F}_i$            `// Parent of` $f_i$

**for** $i = M - 1, M - 2, \ldots, 0$ and $f_i \in \mathcal{F}_i$ **do**

    Let $C(f_i) = \{f_{i+1} \in \mathcal{F}_{i+1} : f_i = P(f_{i+1})\}$    `// Children of` $f_i$

    Let $\mathcal{L}(f_i) = \bigcup_{f_{i+1} \in C(f_i)} \mathcal{L}(f_{i+1})$        `// Leaves of` $f_i$

    Let $\mathcal{A}(f_i)$ be an instance of Algorithm 1, $\Delta = 2^{-i+3}$ and
    $K = |C(f_i)| + 1$    `// Algorithm for` $f_i$`, with actions its`
    `children and the good action`

**end**

**for** $t \in [T]$ **do**

    Receive $v_t, \chi_t, \psi_t$ and calculate $U_t = \max_{\tau \leq t} \{\chi_\tau, \psi_\tau\}$   `// `$U_t$` is`
    `the upper bound of rewards`

    For every $f_M \in \mathcal{F}_M$ calculate $b_t^{f_M}$ as in Eq. (14) `// Improve bid`
    $f_M(v_t)$ `using safe bid`

    and let $q_t^{f_M}(\cdot)$ be the bid distribution that bids $b_t^{f_M}$ with
    probability 1               `// `$f_M$`'s suggestion`

    `// Recursive construction of bid distributions`

    **for** $i = M - 1, M - 2, \ldots, 0$ and $f_i \in \mathcal{F}_i$ **do**

        Calculate $g_t^{f_i} = \max_{f_M \in \mathcal{L}(f_i)} b_t^{f_M}$      `// `$f_i$`'s good bid`

        Get $p_t^{f_i}(\cdot)$ from $\mathcal{A}(f_i)$ after passing $U_t$     `// Distribution`
        over $C(f_i) \cup \{g_t^{f_i}\}$

        Calculate bid distribution $q_t^{f_i}(\cdot)$ over bids $\{b_t^{f_M}\}_{f_M \in \mathcal{L}(f_i)}$

        where $q_t^{f_i}\left(b_t^{f_M}\right)$ equals

$$p_t^{f_i}\left(g_t^{f_i}\right) \mathbb{1}\left[g_t^{f_i} = b_t^{f_M}\right] + \sum_{f_{i+i} \in C(f_i)} p_t^{f_i}(f_{i+1}) q_t^{f_{i+1}}\left(b_t^{f_M}\right)$$

    **end**

    Sample and use bid $b_t \sim q_t^{f_0}(\cdot) = q_t(\cdot)$         `// Sample bid`
    `according to the root`

    Receive $d_t$         `// Get `$d_t$`, making `$r_t(\cdot)$` calculable`

    `// Update algorithms of non-leaf nodes`

    **for** $i = 0, 1, \ldots, M$ and $f_i \in \mathcal{F}_i$ **do**

        Let $\tilde{r}_t(g_t^{f_i}) = r_t(g_t^{f_i})$        `// Reward of good bid`

        Let $\tilde{r}_t(f_{i+1}) = \sum_b q_t^{f_{i+1}}(b) r_t(b)$, for every $f_{i+1} \in C_{i+1}$
        `// Reward of each child of` $f_i$

        Pass above $\tilde{r}_t(\cdot)$ to $\mathcal{A}(f_i)$     `// Update` $f_i$`'s algorithm`

    **end**

**end**

---

We now use Algorithm 2 to prove Theorem 3.6.

PROOF OF THEOREM 3.6. Fix $M = \lceil \log_2 \sqrt{T} \rceil$. We first make the observation that Algorithm 2 is well defined: in order to calculate $b_t^{f_M}$ for every $f_M \in \mathcal{F}_M$ we only need knowledge of $v_t, \chi_t, \psi_t$ and not $d_t$, as explained in Assumption 3.1. We also note that the rewards

$\tilde{r}_t(\cdot)$ that are fed into each $\mathcal{A}(f_i)$ are in the range $[0, U_t]$ and $U_1 \le U_2 \le \ldots$, as needed for the guarantee of Theorem 3.5. The lower bound for the rewards comes from the fact that every bid used in round $t$ is $b_t^{f_M}$ for some $f_M \in \mathcal{F}_M$, which because of (14) and Assumption 3.1 guarantees non-negative reward. The upper bound follows from the definition of $r_t(\cdot)$ and the fact that values and bids are in $[0, 1]$.

We now show that for every $i < M$ and $f_i \in \mathcal{F}_i$, the good bid, $g_t^{f_i}$, is $2^{-i+3}$-good with respect to the bids used by $f_i$, i.e. $\{b_t^{f_M}\}_{f_M \in \mathcal{L}(f_i)}$. We prove that

$$r_t\left(g_t^{f_i}\right) \ge r_t\left(b_t^{f_M}\right) - 2^{-i+3}U_t, \quad \forall f_M \in \mathcal{L}(f_i) \qquad (15)$$

which implies $\tilde{r}_t(g_t^{f_i}) \ge \tilde{r}_t(f_{i+1}) - 2^{-i+3}U_t$ for all $f_{i+1} \in C(f_i)$, since $\tilde{r}_t(g_t^{f_i}) = r_t(g_t^{f_i})$ and $\tilde{r}_t(f_{i+1})$ is a convex combination of $\{r_t(b_t^{f_M})\}_{f_M \in \mathcal{L}(f_i)}$.

Fix $f_M \in \mathcal{L}(f_i)$. We distinguish two cases to prove (15) for this $f_M$:

- If $b_t^{f_M}$ loses the auction in round $t$ ($b_t^{f_M} < d_t$), then $r_t(b_t^{f_M}) = 0$ and (15) follows from $r_t(g_t^{f_i}) \ge 0$.

- If $b_t^{f_M}$ wins the auction in round $t$, then $g_t^{f_i} \ge b_t^{f_M}$ and therefore $g_t^{f_i}$ also wins the auction in round $t$. This means that $\mathbb{1}\left[b_t^{f_M} \ge d_t\right] = \mathbb{1}\left[g_t^{f_i} \ge d_t\right] = 1$ and so, in order to prove (15) we have to prove that the payment of $g_t^{f_i}$ is not more than the payment of $b_t^{f_M}$ plus $2^{-i+3}$. The last statement follows from the Lipschitzness of $p(\cdot, d_t)$ (Assumption 3.1) and the fact that $|g_t^{f_i} - b_t^{f_M}| \le 2^{-i+3}$ which follows by the following: For every $f_M, f'_M \in \mathcal{L}(f_i)$ it holds that $\|f_M - f'_M\|_\infty \le 2^{-i+3}$ since

$$\|f_i - f_M\|_\infty \le \sum_{j=0}^{M-i-1} \left\|P^{j+1}(f_M) - P^j(f_M)\right\|_\infty$$

$$\le \sum_{j=0}^{M-i-1} 2^{-M+j+2} \le 2^{-i+2}$$

where $P^j(\cdot)$ is the application of the parent function $P$ $j$ times, the first inequality uses the triangle inequality, and the second uses the definition of the the parent function $P$. The fact that $\|f_M - f'_M\|_\infty \le 2^{-i+3}$ follows from the above using the triangle inequality and the fact that $|g_t^{f_i} - b_t^{f_M}| \le 2^{-i+3}$ follows from the fact that $g_t^{f_i} = b_t^{f'_M}$ for some $f'_M \in \mathcal{L}(f_i)$.

Now we summarize the setting of each algorithm $\mathcal{A}(f_i)$, for $f_i \in \mathcal{F}_i, i < M$:

- The reward range is $[0, U_t]$ in round $t$, where $U_t = \max\{\chi_t, \psi_t\}$.
- In every round there is an action that is $\Delta_i$-good, where $\Delta_i := 2^{-i+3}$.
- There are at most $K_i$ actions, where $K_i := \exp(C_{\text{Lip}}L2^{i+1}) + 1 \le \exp(C_{\text{Lip}}L2^{i+2})$ where last inequality holds because $C_{\text{Lip}}L \ge 1$.

Let $\tilde{R}_T(f_i) = \sum_{t=1}^T \tilde{r}_t(f_i)$ denote the total reward of algorithm $\mathcal{A}(f_i)$ and similarly define $\tilde{R}_T(g^{f_i})$ the total reward of the good bids of $\mathcal{A}(f_i)$. Because of the guarantee of each algorithm (Theorem 3.5)

we have that with probability 1:

$$\max_{f_{i+1} \in C(f_i) \cup \{g^{f_i}\}} \tilde{R}_T(f_{i+1}) - \sum_{f_{i+1} \in C(f_i) \cup \{g^{f_i}\}} p_t^{f_i}(f_{i+1}) \tilde{R}_T(f_{i+1})$$

$$\le 4U\sqrt{T\Delta_i \log K_i} \le 23U\sqrt{C_{\text{Lip}}LT} \qquad (16)$$

We now bound the error because we bid according to the bidding functions $\mathcal{F}_M$ and not $\mathcal{F}$. For any $f \in \mathcal{F}$ let $f_M \in \mathcal{F}_M$ be such that $f_M \ge f$ and $\|f - f_M\|_\infty \le 2^{-M}$. For every round $t$ we have

$$r_t(f)$$

$$= \mathbb{1}[f(v_t) \ge d_t]\left(\chi_t v_t - \psi_t p(f(v_t), d_t)\right)$$

$$\le \mathbb{1}[f_M(v_t) \ge d_t]\left(\chi_t v_t - \psi_t p(f(v_t), d_t)\right)^+$$

$$\le \mathbb{1}[f_M(v_t) \ge d_t]\left(\chi_t v_t - \psi_t p(f_M(v_t), d_t)\right)^+ + \psi_t f_M(v_t) - \psi_t f(v_t)$$

$$\le \mathbb{1}[f_M(v_t) \ge d_t]\left(\chi_t v_t - \psi_t p(f_M(v_t), d_t)\right)^+ + U_t 2^{-M}$$

$$= \left(r_t(f_M)\right)^+ + U_t 2^{-M} \le \tilde{r}_t(f_M) + U_t 2^{-M}$$

where in the first inequality we used that $f_M \ge f$, in the second inequality that $p(\cdot, d_t)$ is 1-Lipschitz, in the third that $\|f - f_M\|_\infty \le 2^{-M}$, and in the final one that $\tilde{r}_t(f_M) \ge (r_t(f_M))^+$; recall that $\tilde{r}_t(f_M) = r_t(b_t^{f_M})$.

The above implies

$$\sup_{f \in \mathcal{F}} \sum_{t=1}^T r_t(f) \le \max_{f_M \in \mathcal{F}_M} \tilde{R}_T(f_M) + UT 2^{-M} \qquad (17)$$

Let $f_M^*$ be a maximizer of the r.h.s. in the inequality above and for every $i < M$, let $f_i^*$ be the ancestor of $f_M^*$ in the $i$-th level. Using this notation we prove

$$\sup_{f \in \mathcal{F}} \sum_{t=1}^T r_t(f) - \tilde{R}_T(f_0) \le UT 2^{-M} + \tilde{R}_T(f_M^*) - \tilde{R}_t(f_0)$$

$$= UT 2^{-M} + \sum_{j=0}^{M-1} \left(\tilde{R}_T(f_{j+1}^*) - \tilde{R}_T(f_j^*)\right)$$

$$\le UT 2^{-M} + \sum_{j=0}^{M-1} 23U\sqrt{C_{\text{Lip}}LT}$$

$$= UT 2^{-M} + 23UM\sqrt{C_{\text{Lip}}LT}$$

where the first inequality holds by (17) and the second one by (16). Picking $M = \lfloor \log_2 \sqrt{T} \rfloor$ the above becomes

$$\sup_{f \in \mathcal{F}} \sum_{t=1}^T r_t(f) - \tilde{R}_T(f_0) \le 2U\sqrt{T} + \frac{23}{2\log 2}U\sqrt{C_{\text{Lip}}LT}\log T$$

The above is the claimed regret bound since $\tilde{r}_t(f_0) = \sum_b q_t^{f_0} r_t(b)$. $\qquad \square$

## D.4 Deferred Proofs and Algorithm of Section 3.4

We first present in Algorithm 3 the reduction from standard no-regret to no interval regret. We use that to prove Theorem 3.7.

---

**ALGORITHM 3:** Reduction from regret to interval regret

---

**Input:** Number of rounds $T$, action space $A$, algorithms $\left\{\mathcal{A}_{\tau_1}\right\}_{\tau_1 \in [T]}$
over action space $A$

Initialize an instance $\mathcal{A}$ of Algorithm 1 with $\Delta = 1$ and $K = T$

**for** $t \in [T]$ **do**

    Receive reward range $[0, U_t]$

    **for** $\tau_1 \leq t$ **do**

        Pass $U_t$ to $\mathcal{A}_{\tau_1}$ and receive $q_t^{\tau_1}(\cdot)$    // Distribution over
        actions of $\mathcal{A}_{\tau_1}$

    **end**

    Pass $U_t$ to $\mathcal{A}$ and receive $p_t(\cdot)$    // Distribution over
    algorithms of $\mathcal{A}$

    Calculate for every action $a$: $q_t(a) = \frac{\sum_{\tau_1 \leq t} p_t(\tau_1) q_t^{\tau_1}(a)}{\sum_{\tau_1 \leq t} p_t(\tau_1)}$
    // Distribution of actions by sampling an algorithm
    $\mathcal{A}_{\tau_1}, t \leq \tau_1$ and then an action from $q_t^{\tau_1}(\cdot)$

    Sample and output $a_t \sim q_t(\cdot)$

    Receive function $r_t : A \to [0, U_t]$

    Pass $r_t(\cdot)$ to $\mathcal{A}_{\tau_1}$ for $\tau_1 \leq t$    // $\mathcal{A}_{\tau_1}$ internal update

    Calculate $\tilde{r}_t(\tau_1) = \mathbb{E}_{a \sim q_t^{\tau_1}}[r_t(a)]$ for $\tau_1 \leq t$    // Expected
    reward of $q_t^{\tau_1}(\cdot)$

    Calculate $\tilde{r}_t^{\emptyset} = \mathbb{E}_{a \sim q_t}[r_t(a)]$    // Expected reward of $q_t(\cdot)$

    Update $\mathcal{A}$ with reward $\tilde{r}_t(\tau_1)$ for $\tau_1 \leq t$    // $\mathcal{A}$ update for
    active algorithms
    and with reward $\tilde{r}_t^{\emptyset}$ for $\tau_1 \leq t$    // $\mathcal{A}$ update for inactive
    algorithms

**end**

---

PROOF OF THEOREM 3.7. We first extend the definition of $\tilde{r}_t(\cdot)$ for inactive algorithms. This makes

$$\tilde{r}_t(\tau_1) = \begin{cases} \mathbb{E}_{a \sim q_t^{\tau_1}}[r_t(a)], & \text{if } \tau_1 \leq t \\ \tilde{r}_t^{\emptyset}, & \text{if } \tau_1 > t \end{cases}.$$

Now we re-write $\tilde{r}_t^{\emptyset}$, the expected reward if the action is sampled according to $q_t(\cdot)$:

$$\tilde{r}_t^{\emptyset} = \mathbb{E}_{a \sim q_t}[r_t(a)] = \frac{1}{\sum_{\tau_1 \leq t} p_t(\tau_1)} \sum_{\tau_1 \leq t} \left(p_t(\tau_1)\tilde{r}_t(\tau_1)\right). \quad (18)$$

We notice that $\tilde{r}_t(\cdot)$ has the same reward range as $r_t(\cdot)$. Theorem 3.5 for algorithm $\mathcal{A}$ gives us a regret guarantee by every round $\tau_2$:

$$\max_{\tau_1 \in [T]} \sum_{t \in [\tau_2]} \tilde{r}_t(\tau_1) - \sum_{t \in [\tau_2]} \sum_{\tau_1 \in [T]} p_t(\tau_1)\tilde{r}_t(\tau_1) \leq 4U_{\tau_2}\sqrt{T \log T}$$

which implies that for all intervals $[\tau_1, \tau_2] \subseteq [T]$

$$\sum_{t \in [\tau_2]} \tilde{r}_t(\tau_1) - \sum_{t \in [\tau_2]} \sum_{\tau_1 \in [T]} p_t(\tau_1)\tilde{r}_t(\tau_1) \leq 4U_{\tau_2}\sqrt{T \log T} \quad (19)$$

We are going to show that (19) implies our theorem. First we prove that for every round $t$,

$$\sum_{\tau_1 \in [T]} p_t(\tau_1)\tilde{r}_t(\tau_1) = \sum_{\tau_1 \leq t} p_t(\tau_1)\tilde{r}_t(\tau_1) + \sum_{\tau_1 > t} p_t(\tau_1)\tilde{r}_t^{\emptyset}$$

$$= \tilde{r}_t^{\emptyset} \sum_{\tau_1 \leq t} p_t(\tau_1) + \tilde{r}_t^{\emptyset} \sum_{\tau_1 > t} p_t(\tau_1)$$

$$= \tilde{r}_t^{\emptyset}$$

where the second equality holds by (18).

The above and the fact that $\tilde{r}_t(\tau_1) = \tilde{r}_t^{\emptyset}$ for $\tau_1 > t$ makes (19) imply that for all intervals $[\tau_1, \tau_2] \subseteq [T]$

$$\sum_{t < \tau_1} \tilde{r}_t(\emptyset) + \sum_{t \in [\tau_1, \tau_2]} \tilde{r}_t(\tau_1) - \sum_{t \in [\tau_2]} \tilde{r}_t(\emptyset) \leq 4U_{\tau_2}\sqrt{T \log T}$$

or equivalently that for all such intervals $[\tau_1, \tau_2] \subseteq [T]$

$$\sum_{t \in [\tau_1, \tau_2]} \tilde{r}_t(\tau_1) - \sum_{t \in [\tau_1, \tau_2]} \tilde{r}_t(\emptyset) \leq 4U_{\tau_2}\sqrt{T \log T}.$$

Given the regret bound of each algorithm $\mathcal{A}_{\tau_1}$ by round $\tau_2$ the above implies that for all intervals $[\tau_1, \tau_2] \subseteq [T]$

$$\sup_a \sum_{t \in [\tau_1, \tau_2]} r_t(a) - \sum_{t \in [\tau_1, \tau_2]} \tilde{r}_t(\emptyset) \leq 4U_{\tau_2}\sqrt{T \log T} + \text{Reg}_{\tau_1}(\tau_2).$$

which is the desired regret bound. □

Using a simple concentration inequality and the union bound, we prove Theorem 3.2 from Corollary 3.8.

PROOF OF THEOREM 3.2. Fix $1 \leq \tau_1 < \tau_2 \leq T$. For $t \in [\tau_1, \tau_2]$, define $X_t = r_t(b_t) - \sum_b q_t(b)r_t(b)$ and $M_t = \sum_{t' \in [\tau_1, t]} X_t$. We notice that the sequence $M_t$ is a martingale with respect to the history of the previous rounds $\mathcal{H}_{t-1}$ (player's and adversary's decisions): for every $t$

$$\mathbb{E}\left[M_t - M_{t-1} \mid \mathcal{H}_{t-1}\right] = \mathbb{E}\left[X_t \mid \mathcal{H}_{t-1}\right] = 0$$

In addition we notice that $|X_t| \leq U_t$ since $r_t(b) \in [0, U_t]$. This allows us to use Azuma's inequality, proving that for every $\delta \in [0, 1]$, with probability at least $1 - \delta$ it holds

$$M_{\tau_2} \geq -U_{\tau_2}\sqrt{2(\tau_2 - \tau_1)\log(1/\delta)}$$

which implies that with probability at least $1 - \delta$

$$\sum_{t=\tau_1}^{\tau_2} r_t(b_t) - \sum_{t=\tau_1}^{\tau_2} \sum_b q_t(b)r_t(b) \geq -U_{\tau_2}\sqrt{2T \log(1/\delta)}$$

Using the union bound over all $1 \leq \tau_1 < \tau_2 \leq T$ we get that for every $\delta \in [0, 1]$ with probability at least $1 - \delta$ it holds that for all $1 \leq \tau_1 < \tau_2 \leq T$

$$\sum_{t=\tau_1}^{\tau_2} r_t(b_t) - \sum_{t=\tau_1}^{\tau_2} \sum_b q_t(b)r_t(b) \geq -U_{\tau_2}\sqrt{2T \log\frac{\binom{T}{2}}{\delta}}$$

Using Corollary 3.8 we get the theorem. □

# E Deferred Proofs of Section 4

We first prove the reduction of how to turn approximate ROI satisfaction to an exact one.

PROOF OF LEMMA 4.2. We first notice that since $\mathcal{A}_2$ never has value less than payment and $\mathcal{A}_1$ is run only when the accumulated value is at least 1 higher than the payment, the ROI constraint is never going to be violated. Now we need to prove the total value guarantee.

Assume that the high probability bounds of the two algorithms are true (which happens with probability at least $1 - 2\delta$ for any $\delta$ due to the union bound). Let $\tau$ be the last round when algorithm $\mathcal{A}_2$ is run. Let $\mathcal{T}_1$ be the rounds up to $\tau$ where algorithm $\mathcal{A}_1$ is run and $\mathcal{T}_2$ be the rounds up to $\tau$ where algorithm $\mathcal{A}_2$ is run; note that $|\mathcal{T}_2|$ is the total number of rounds $\mathcal{A}_2$ is run in total. We now have

$$\sum_{t \in [\tau]} \mathbb{1}\left[b_t \geq d_t\right](v_t - p(b_t, d_t)) \geq Q_\delta(|\mathcal{T}_2|) - V_\delta(|\mathcal{T}_1|)$$

where the inequality follows from the ROI guarantees of the two algorithms. Using the fact that on round $\tau$ we run $\mathcal{A}_2$ which upper bounds the above quantity by 2 we get

$$Q_\delta(|\mathcal{T}_2|) \le 2 + V_\delta(|\mathcal{T}_1|) \le 2 + V_\delta(T)$$

Using the definition that $Q_\delta^{-1}(V_\delta(T,))$ is the solution to $Q_\delta(\cdot) = V_\delta(T)$ we get

$$|\mathcal{T}_2| \le Q_\delta^{-1}(V_\delta(T))$$

This proves that the total number of rounds $\mathcal{A}_1$ was run is at least $T - Q_\delta^{-1}(1 + V(T))$. This means that the total regret is at most

$$\mathsf{Reg}_\delta\left(T - Q_\delta^{-1}(1 + V(T))\right) + 2Q_\delta^{-1}(1 + V_\delta(T))$$

where the second term represents the rounds $\mathcal{A}_2$ was run instead of $\mathcal{A}_1$ and the loss because of the budget consumption of $\mathcal{A}_2$, which is at most $Q_\delta^{-1}(V_\delta(T))$ making the overall algorithm run out of budget $Q_\delta^{-1}(V_\delta(T))$ rounds earlier, missing out on that much utility. □

We now prove Theorem 4.1.

PROOF OF THEOREM 4.1. As we explained in Section 4 there are algorithms that satisfy the assumptions of Lemma 4.2 with

$$\mathsf{Reg}_\delta(T) = V_\delta(T) = \frac{1}{\rho\beta} O\left(\sqrt{LT}\log T + \sqrt{T\log(1/\delta)}\right)$$

and

$$Q_\delta(T_2) = \beta T_2 - O\left(\sqrt{LT_2}\log T_2 + \sqrt{T_2\log(T_2/\delta)}\right) = \Omega(\beta T_2)$$

where the last inequality follows from

$$\beta = \omega\left(\sqrt{\frac{L}{T}}\log T + \sqrt{\frac{\log(T/\delta)}{T}}\right)$$

since otherwise the regret statement is vacuous.

The theorem follows by algebraic calculations. □

We now prove Theorem 4.3.

PROOF OF THEOREM 4.3. Each round the value/highest-competing-bid distribution is the following

$$(v_t, d_t) = \begin{cases} (1, 0), & \text{w.p. } \beta \\ \left(\frac{1-2\beta}{1-\beta}, 1\right), & \text{w.p. } 1 - \beta \end{cases}$$

It is not hard to see that the LP optimum bids 1 every round and on expectation has reward $\mathsf{OPT} = 1 - \beta$ and ROI violation of 0.

Let $R_t$ be the cumulative "ROI amount" that any algorithm has collected in round $t$, i.e. the total value minus the total price paid. Because the algorithm needs to satisfy the ROI constraint with probability 1, for all rounds it must hold that $R_t \ge 0$. This means that the best any algorithm can do is to bid 1 in round $t$ if $R_{t-1} \ge \frac{\beta}{1-\beta}$ (thus guaranteeing to always win) or bid less than 1 if $R_{t-1} < \frac{\beta}{1-\beta}$ (thus guaranteeing to win only if $d_t = 0$).

Let $N_t$ be the number of rounds the algorithm bid less than 1 and missed an item with $d_t = 1$. The regret of the algorithm is $N_T \frac{1-2\beta}{1-\beta}$ so we need to calculate $\mathbb{E}[N_T]$.

Let $R_t'$ be the cumulative "ROI amount" if the algorithm won every item. We notice that

$$R_t - R_t' = \frac{\beta}{1-\beta}N_t$$

since the difference does not change from round to round if the algorithm wins the item but increases if the the algorithms misses a high value item. Since $R_t \ge 0$ and it holds that

$$N_T = \min_t N_t \ge -\frac{1-\beta}{\beta}\min_t R_t'$$

We now note that $R_t'$ is an unbiased random walk, since

$$R_t' - R_{t-1}' = \begin{cases} 1, & \text{w.p. } \beta \\ \frac{\beta}{1-\beta}, & \text{w.p. } 1 - \beta \end{cases}$$

and $\mathbb{E}[R_t' - R_{t-1}'] = 0$ and $\mathrm{Var}[R_t' - R_{t-1}'] = \frac{\beta}{1-\beta}$. We will show that $\mathbb{E}\left[\min_t R_t'\right] \le -\sqrt{T\frac{\beta}{2\pi(1-\beta)}}$, which in turn implies that

$$\mathbb{E}[N_T] \ge \frac{1-\beta}{\beta}\sqrt{T\frac{\beta}{2\pi(1-\beta)}} = \sqrt{T\frac{1-\beta}{2\pi\beta}}$$

which implies that the expected regret of the algorithm wrt to the LP optimum is

$$(1 - 2\beta)\sqrt{T\frac{1}{2\pi\beta(1-\beta)}}$$

which proves the theorem.

We now prove that $\mathbb{E}\left[\min_t R_t'\right] \le -\sqrt{T\frac{\beta}{2\pi(1-\beta)}}s$. We have that

$$\begin{aligned} \mathbb{E}\left[\min_{t=0,\dots,T} R_t'\right] &= \mathbb{E}\left[\min_{t=0,\dots,T}\min\{0, R_t'\}\right] \\ &\le \mathbb{E}\left[\min\{0, R_T'\}\right] \\ &= -\frac{1}{2}\mathbb{E}\left[|R_T'|\right] \\ &\approx -\frac{1}{2}\mathbb{E}\left[\left|G\left(0, \sqrt{T\frac{\beta}{1-\beta}}\right)\right|\right] \\ &= -\frac{1}{2}\sqrt{T\frac{\beta}{1-\beta}}\sqrt{\frac{2}{\pi}} \end{aligned}$$

where the second equality holds because $\mathbb{E}\left[R_T'\right] = \mathbb{E}\left[\max\{0, R_T'\}\right] + \mathbb{E}\left[\min\{0, R_T'\}\right] = 0$ and $\mathbb{E}\left[|R_T'|\right] = \mathbb{E}\left[\max\{0, R_T'\}\right] - \mathbb{E}\left[\min\{0, R_T'\}\right]$, the next one by the Central Limit Theorem for large $T$, and the last equality uses standard facts of $G\left(0, \sqrt{T\frac{\beta}{1-\beta}}\right)$, the 0-mean Gaussian with standard deviation $\sqrt{T\frac{\beta}{1-\beta}}$.

We note that instead of using the central limit theorem one could explicitly use the fact that $R_T' = M\left(1 + \frac{\beta}{1-\beta}\right) - T\frac{\beta}{1-\beta}$ where $M$ is a binomial random variable with $T$ tries and probability of success $b$ to calculate

$$\begin{aligned} \mathbb{E}\left[|R_T'|\right] &= \left(1 + \frac{\beta}{1-\beta}\right)\mathbb{E}\left[|M - T\beta|\right] \\ &= \left(1 + \frac{\beta}{1-\beta}\right)2\lceil T\beta\rceil(1-\beta)^{T+1-\lceil T\beta\rceil}\beta^{\lceil T\beta\rceil}\binom{T}{\lceil T\beta\rceil} \\ &= \sqrt{\frac{2\beta}{\pi(1-\beta)}}\sqrt{T} - O\left(\frac{1}{\sqrt{T}}\right) \end{aligned}$$

which leads to a similar bound. □

# F Deferred Proofs and Algorithm of Section 5

We first present in Appendix F.1 the deferred proof of Theorem F.4, the lower bound on the regret for budgeted first-price auctions with only bandit information. Next we present an upper bound on the regret, in Appendix F.2.

## F.1 Deferred Proof for Regret Lower Bound

In this section we prove Theorem 5.1.

PROOF OF THEOREM 5.1. Fix the player's budget, $\rho = 1/4$. Define the following

- $K = 2T^{1/3}$ for some positive $c$ that will be defined later.
- $\varepsilon = \frac{1}{3K}$.
- For $i = 0, 1, 2, \ldots, K$ let $d_i = 1/3 + i\varepsilon$. Note $d_0 = 1/3$ and $d_K = 2/3$.

We will consider that $d_t$ can only be 0, 1, or one of the values in $\{d_i\}_{i=0,\ldots,K-1}$; note that even though we defined the value $d_K$, $d_t$ cannot take that value. We will consider different distributions that can generate $d_t$, each specified by a CDF. We first define the base CDF-like distribution:

$$F_b(x) = \frac{3}{4-x} \tag{20}$$

We use $\mathbb{P}_b$ and $\mathbb{E}_b$ to denote the probability and expectation when $d_t$ is generated by $F_b$, which means that for any $x = 0, d_0, d_1, \ldots, d_{K-1}$, $\mathbb{P}_b[d_t \le x] = F_b(x)$ and $\mathbb{P}_b[d_t \le 1] = 1$. More precisely, even though we do not use the description of the probability density function, we have

$$\mathbb{P}_b[d_t = x] = \begin{cases} F_b(0) = 3/4 & \text{, if } x = 0 \\ F_b(d_0) - F_b(0) = 3/44 & \text{, if } x = d_0 = 1/3 \\ F_b(d_i) - F_b(d_{i-1}) = \Theta(\varepsilon) & \text{, if } x = d_i, i \in [K-1] \\ 1 - F_b(d_{K-1}) & \text{, if } x = 1 \\ 0 & \text{, otherwise} \end{cases}$$

We note that $F_b(d_{K-1}) < F_b(d_K) = 1/10$; this guarantees that our distribution is well defined.

Given the base CDF $F_b$ we now define a different CDF-like function, $F_j$ for every $j = 0, 1, \ldots, K-1$. The distribution of $d_t$ associated with $F_j$ is going to be identical to the one associated with $F_b$. except for the probability of

$$F_j(x) = \begin{cases} F_b(x), & \text{if } x \notin [d_j, d_{j+1}) \\ F_b(d_{j+1}), & \text{if } x \in [d_j, d_{j+1}) \end{cases} \tag{21}$$

We use $\mathbb{P}_j$ and $\mathbb{E}_j$ to denote the probability and expectation when $d_t$ is generated by $F_j$. This means that $\mathbb{P}_j[d_t \le x] = \mathbb{P}_b[d_t \le x]$ for all $x = 0, d_0, d_1, \ldots, d_{j-1}, d_{j+1}, \ldots, d_{K-1}, 1$ and $\mathbb{P}_j[d_t \le d_j] = \mathbb{P}_b[d_t \le d_{j+1}]$.

We are going to assume that before round $t$, $j$ is picked adversarially. We now prove what a lower bound on the value of the optimal solution under $F_j$.

LEMMA F.1. When $d_t$ is generated according to $F_j$ for any $j = 0, \ldots, K-1$ as explained above, then the value of the optimal solution is

$$\mathsf{OPT}_j \ge \frac{13}{16}T + \frac{3}{32}T\varepsilon$$

PROOF. We consider the strategy that bids $d_j$ with probability $\frac{11-3(i+1)\varepsilon}{12(1+3i\varepsilon)}$ and bids 0 otherwise. The expected per round payment of this strategy is

$$\frac{11-3(i+1)\varepsilon}{12(1+3i\varepsilon)}d_j F_j(d_j) = \frac{11-3(i+1)\varepsilon}{12(1+3i\varepsilon)}\left(\frac{1}{3}+i\varepsilon\right)\frac{3}{4-(\frac{1}{3}+(i+1)\varepsilon)}$$

$$= \frac{1}{4}$$

which means it satisfies the budget constraint in expectation, since $\rho = 1/4$. The expected per-round value of this solution is

$$\frac{11-3(i+1)\varepsilon}{12(1+3i\varepsilon)}F_j(d_j) + \left(1 - \frac{11-3(i+1)\varepsilon}{12(1+3i\varepsilon)}\right)F_j(0)$$

$$= \frac{11-3(i+1)\varepsilon}{12(1+3i\varepsilon)}\frac{3}{4-(\frac{1}{3}+(i+1)\varepsilon)} + \left(1 - \frac{11-3(i+1)\varepsilon}{12(1+3i\varepsilon)}\right)\frac{3}{4}$$

$$= \frac{13}{16} + \frac{3\varepsilon}{16(1+3i\varepsilon)} \ge \frac{13}{16} + \frac{3\varepsilon}{16(1+3K\varepsilon)}$$

where the last inequality follows from $i \le K$. Substituting $\varepsilon = \frac{1}{3K}$ we get the lemma. □

In round $t$ the player bids $b_t$ and observes $x_t \in \{0, 1\}$ (if she won or not). We are going to assume that the player runs some deterministic algorithm, which comes w.l.o.g. since the environment is randomized. This means that the player's bid $b_t$ in round $t$ is a deterministic function of $x_1, \ldots, x_{t-1}$. We denote $x = x_{1:T}$. We also assume w.l.o.g. that the player's bids are always one of $0, d_0, \ldots, d_{K-1}, 1$, since any other bid is suboptimal. Let $N_i$ be the total number of times the player bids $d_i$ and $M_i$ be the total number of times the player bids $d_i$ and wins. We prove the following.

LEMMA F.2. Let $g : \{0, 1\}^T \to \mathbb{R}$ be a function defined on $x$. Then, for every $j$, it holds that

$$\left|\mathbb{E}_j g(x) - \mathbb{E}_b g(x)\right| \le \left(\max_x |g(x)|\right)\varepsilon\sqrt{2\,\mathbb{E}_b[M_j]}$$

PROOF. For any $j$ and $i$ we have

$$\left|\mathbb{E}_j f(x) - \mathbb{E}_b f(x)\right| = \left|\sum_{x \in \{0,1\}^T} f(x)\left(\mathbb{P}_j[x] - \mathbb{P}_b[x]\right)\right|$$

$$\le \left(\max_x |f(x)|\right)\sum_{x \in \{0,1\}^T}\left|\mathbb{P}_j[x] - \mathbb{P}_b[x]\right|$$

$$= \left(\max_x |f(x)|\right)\left|\mu_j - \mu_b\right|_1$$

where $\mu_j$ and $\mu_b$ are the probability distributions on $x$ given that $d_t$ is sampled from $F_j$ and $F_b$, respectively. Using the properties of the KL divergence, we have that

$$\left|\mu_j - \mu_b\right|_1^2 \le 2D_{KL}\left(\mu_b \,\|\, \mu_j\right)$$

Now fix some sequence $x$. Conditioned on $x_{1:t-1}$, $b_t$ is deterministic. In addition, if $b_t \ne d_j$ it holds

$$\mu_b(x_t = 1|x_{1:t-1}) = \mu_j(x_t = 1|x_{1:t-1})$$

which, if $b_t \ne d_j$, implies

$$D_{KL}\left(\mu_b(x_t = \cdot|x_{1:t-1}) \,\|\, \mu_j(x_t = \cdot|x_{1:t-1})\right) = 0$$

When $b_t = d_j$ it holds that

$$\mu_b(x_t = 1|x_{1:t-1}) = F_b(d_j) = \frac{3}{4 - (d_0 + j\varepsilon)}$$

$$\mu_j(x_t = 1|x_{1:t-1}) = F_j(d_j) = F_b(d_{j+1}) = \frac{3}{4 - (d_0 + (j+1)\varepsilon)}$$

which implies that if $b_t = d_j$

$$D_{KL}\left(\mu_b(x_t = \cdot|x_{1:t-1}) \,\middle\|\, \mu_j(x_t = \cdot|x_{1:t-1})\right)$$

$$= D_{KL}\left(\text{Bern}(F_b(d_j)) \,\middle\|\, \text{Bern}(F_b(d_{j+1}))\right)$$

$$= F_b(d_j) \log \frac{F_b(d_j)}{F_b(d_{j+1})} + (1 - F_b(d_j)) \log \frac{1 - F_b(d_j)}{1 - F_b(d_{j+1})}$$

$$\leq F_b(d_j)\left(\frac{F_b(d_j)}{F_b(d_{j+1})} - 1\right) + (1 - F_b(d_j))\left(\frac{1 - F_b(d_j)}{1 - F_b(d_{j+1})} - 1\right)$$

$$= 3\frac{\varepsilon^2}{(1 - d_j)(4 - d_{j+1})^2} \leq \frac{81}{100}\varepsilon^2$$

where $\text{Bern}(p)$ is a Bernoulli random variable with mean $p$, the first inequality follows by using $\log x \leq x - 1$ for all $x > 0$ and the last inequality follows by using $d_j, d_{j+1} \leq 2/3$. This implies that for any $b_t$,

$$D_{KL}\left(\mu_b(x_t = \cdot|x_{1:t-1}) \,\middle\|\, \mu_j(x_t = \cdot|x_{1:t-1})\right) \leq \frac{81}{100}\varepsilon^2 \mathbb{1}\left[b_t(x) = d_j\right]$$

Taking expectations over $x \sim \mu_b$ and adding over $t$ we get

$$D_{KL}\left(\mu_b \,\middle\|\, \mu_j\right) \leq \frac{81}{100}\varepsilon^2 \, \mathbb{E}_b\left[N_j\right] \leq \frac{99}{100}\varepsilon^2 \, \mathbb{E}_b\left[M_j\right] \leq \varepsilon^2 \, \mathbb{E}_b\left[M_j\right]$$

where the last inequality follows from the fact that $\mathbb{E}_b\left[M_j\right] = F_b(d_j) \mathbb{E}_b\left[N_j\right]$ and $F_b(d_j) \geq 9/11$. Combining this with what we proved before we get

$$\left|\mathbb{E}_j f(x) - \mathbb{E}_b f(x)\right| \leq \left(\max_x |f(x)|\right)\varepsilon\sqrt{2 \, \mathbb{E}_b\left[M_j\right]}$$

which proves the claim. $\qquad\square$

Now note that since the agent's payment is at least $d_j M_j$, it must always hold $d_j M_j \leq T\rho$, implying $M_j \leq T\rho/d_j \leq \frac{3}{4}T$. Applying Lemma F.2 for $f(x) = M_j$ we get that for any $j$

$$\mathbb{E}_j M_j \leq \mathbb{E}_b M_j + \frac{3\sqrt{2}}{4}T\varepsilon\sqrt{\mathbb{E}_b M_j}$$

We notice that because it must hold $\sum_j d_j M_j \leq T\rho$, there must be some $j$ such that $\mathbb{E}_b M_j \leq \frac{3}{4K}T$. Fix that $j$ and use that in the above inequality, making the above inequality for that $j$

$$\mathbb{E}_j M_j \leq \frac{3T}{4K} + \frac{3\sqrt{2}}{4}T\varepsilon\sqrt{\frac{3T}{4K}} = \frac{3}{4}\frac{T}{K} + \frac{3\sqrt{6}}{8}\frac{T^{3/2}\varepsilon}{\sqrt{K}}$$

Recalling that $K = 2T^{1/3}$ and $\varepsilon = 1/(3K)$ we get

$$\mathbb{E}_j M_j \leq \frac{3}{8}T^{2/3} + \frac{\sqrt{3}}{16}T$$

Since $\mathbb{E}_j M_j = F_j(d_j) \mathbb{E}_j N_j$ and $F_j(d_j) = \frac{3}{4-d_{j+1}} \geq 9/11$ (which follows from $d_{j+1} \geq 1/3$) the above becomes

$$\mathbb{E}_j N_j \leq \frac{33}{56}T^{2/3} + \frac{11\sqrt{3}}{144}T := UT \qquad (22)$$

Using the above upper bound on $\mathbb{E}_j N_j$, we are going to upper bound the value the player can earn in the $j$-th instance. To do that, we are going to consider that the player knows that she is playing against the distribution of $\mathbb{P}_j[\cdot]$ but is restricted by (22). In addition, we consider that the player can satisfy the budget constraint in expectation, since this only increases the budget she can earn. Given

this setting, the player's value is upper bounded by the following LP, where $n_i$ represents the expected number of times the player bids $d_i$, $\ell_0$ is the expected number of times the player bids 0, and $\ell_1$ is the expected number of times the player bids 1:

$$\max_{\ell_0, \ell_1, n_0, \ldots, n_{K-1}} F_j(0)\ell_0 + \ell_1 + \sum_{i=0}^{K-1} F_j(d_i)n_i$$

$$\ell_1 + \sum_{i=0}^{K-1} d_i F_j(d_i)n_i \leq \frac{T}{4}$$

$$\ell_0 + \ell_1 + \sum_{i=0}^{K-1} n_i \leq T$$

$$n_j \leq UT$$

We upper bound the value of the above using its dual:

$$\min_{\lambda, \mu, \nu} \quad \lambda\frac{T}{4} + \mu T + \nu UT$$

$$\mu \geq F_j(0)$$

$$\lambda + \mu \geq 1$$

$$\lambda d_i F_j(d_i) + \mu \geq F_j(d_i), \quad \forall i \neq j$$

$$\lambda d_j F_j(d_j) + \mu + \nu \geq F_j(d_j)$$

We notice that the solution $\lambda = 1/4$, $\mu = 3/4$, $\nu = \varepsilon/4$ is feasible (which follows from some simple but lengthy algebra). This means that the player's expected reward is at most

$$T\left(\frac{13}{16} + \frac{U\varepsilon}{4}\right)$$

Using Lemma F.1 we get that the expected regret it at least

$$\text{Reg} \geq \frac{3}{32}T\varepsilon - \frac{\varepsilon}{4}\left(\frac{33}{56}T^{2/3} + \frac{11\sqrt{3}}{144}T\right)$$

$$= \frac{1}{64}T^{2/3} - \frac{11}{448}T^{1/3} - \frac{11\sqrt{3}}{3456}T^{2/3} \qquad \left(\varepsilon = \frac{T^{-1/3}}{6}\right)$$

$$\approx 0.01T^{2/3} - 0.02T^{1/3}$$

which is $\Omega(T^{2/3})$, as promised. $\qquad\square$

## F.2 Regret Upper bound for Bandit information

In this section we present and prove an upper bound on the regret when learning with bandit feedback. Our regret bound is in the order of $\tilde{O}(T^{3/4})$, which is slightly higher than the lower bound in Theorem 5.1. It remains to be interesting open question to achieve the optimal $O(T^{2/3})$ regret bound.

THEOREM F.3. *There is a polynomial time algorithm for value or quasi-linear utility maximization under Assumption 3.1 such that, for every $\delta > 0$ with probability at least $1 - \delta$ its regret and ROI violation is at most*

$$\frac{1}{\beta\rho}O\left(\left(L^{1/4}T^{3/4} + T^{1/2}L^{1/2}\right)\log\frac{LT}{\delta}\right).$$

*In addition, if $\beta = \Omega\left(T^{-1/4+\varepsilon}(L^{1/4} + L^{1/2}T^{-1/4})\sqrt{\log(TL/d)}\right)$ for some constant $\varepsilon > 0$, the above can be turned in an algorithm with exact ROI satisfaction regret that is $1/\beta$ times worse.*

Theorem F.3 is based on the following theorem, Theorem F.4, which offers an algorithm with $\tilde{O}(\sqrt{TK})$ high probability interval regret bound for the Lagrangian.

---

**ALGORITHM 4:** No interval regret algorithm for bandit information with for time-varying ranges

---

**Input:** Number of rounds $T$, number of actions $K$

Set $\sigma = \frac{1}{T}$, $\xi = \frac{1}{2\sqrt{TK}}$, $\theta = \frac{1}{\sqrt{TK}}$

Initialize weight for each action $w_1(a) = 1 \quad \forall a \in [K]$

**for** $t \in [T]$ **do**

$\quad$ Calculate $p_t(a) = \frac{w_t(a)}{\sum_{a'} w_t(a)}$

$\quad$ Sample and play $a_t \sim p_t(\cdot)$

$\quad$ Receive $\ell_t(a_t)$

$\quad$ Calculate $\tilde{\ell}_t(a) = \frac{\ell_t(a)}{p_t(a)+\xi}\mathbb{1}\left[a = a_t\right]$ for all $a \in [K]$

$\quad$ Receive loss range $[0, U_{t+1}]$ and calculate $\eta_{t+1} = \frac{\theta}{U_{t+1}}$

$\quad$ Calculate $w_{t+1}(a) =$
$\quad\quad (1-\sigma)w_t(a)\exp\left(-\eta_{t+1}\tilde{\ell}_t(a)\right) + \frac{\sigma}{K}\sum_{a'} w_t(a')\exp\left(-\eta_{t+1}\tilde{\ell}_t(a')\right)$

**end**

---

To use this algorithm, we discretize the interval $[0, 1]$ into $N$ values $\{i/N\}_{i\in[N]}$ and $K$ bids $\{j/K\}_{j\in[K]}$. Then for each $i \in [N]$ we run an instance of the algorithm of Theorem F.4, which is used for round $t$ when $v_t \leq \tilde{i}/N < v_t + 1/N$ and outputs a bid $j_t/K$. As in Section 3, we do not directly use the bid suggested by these algorithms, since it might lead to a negative reward in $r_t(\cdot)$; we use the safe bid of Assumption 3.1 when it is possible to get a negative reward. This roughly leads to a total regret bound of $\tilde{O}(\sqrt{TNK})$, along with a discretization error of $O(T(L/N + 1/K))$. Appropriately picking $N, K$ gives the regret bound.

Theorem F.4 provides the $U_{\tau_2}\tilde{O}(\sqrt{TK})$ interval regret bound for every interval $[\tau_1, \tau_2]$. The algorithm used is similar to the EXP-SIX algorithm that [27] uses to bound the regret of the best sequence of actions and that [10] use to bound the interval regret. However, our algorithm is not the same as EXP-SIX: we modify the algorithm to get a regret bound that scales linearly with $U_{\tau_2}$ instead of $U_{\tau_2}^2$ as shown in [10].

THEOREM F.4. *Suppose there are $K$ actions and the reward of round $t$, $r_t : [K] \to [0, U_t]$, is picked by an adaptive adversary. There exists an algorithm that generates actions $a_1, \ldots, a_T$ such that for every $\delta > 0$, with probability at least $1 - \delta$, we have that for all $1 \leq \tau_1 < \tau_2 \leq T$*

$$\max_{a\in[K]}\sum_{t=\tau_1}^{\tau_2} r_t(a) - \sum_{t=\tau_1}^{\tau_2} r_t(a_t) \leq O\left(U_{\tau_2}\cdot\left(\sqrt{TK}+K\right)\log\frac{TK}{\delta}\right)$$

We first present the algorithm for Theorem F.4 which low interval regret. The algorithm is a modification of the algorithm of [27] so that it works for time-varying ranges. We present the algorithm in Algorithm 4.

We now prove Theorem F.4.

PROOF OF THEOREM F.4. First we define

$$p'_{t+1}(a) = \frac{p_t(a)\exp\left(-\eta_t\tilde{\ell}_t(a)\right)}{\sum_{a'} p_t(a')\exp\left(-\eta_t\tilde{\ell}_t(a')\right)}$$

Next we show that for every $a^*$

$$\sum_a p_t(a)\tilde{\ell}_t(a) - \tilde{\ell}_t(a^*) \leq \frac{1}{\eta_t}\log\left(\frac{p'_{t+1}(a^*)}{p_t(a^*)}\right) + \frac{\eta_t}{2}\sum_a p_t(a)\tilde{\ell}_t^2(a) \tag{23}$$

First we notice that

$$\sum_a p_t(a)\exp\left(-\eta_t\tilde{\ell}_t(a)\right)$$

$$\leq 1 - \eta_t\sum_a p_t(a)\tilde{\ell}_t(a) + \frac{\eta_t^2}{2}\sum_a p_t(a)\tilde{\ell}_t^2(a) \qquad \left(\substack{x \geq 0 \implies \\ e^{-x} \leq 1-x+x^2/2}\right)$$

$$\leq \exp\left(-\eta_t\sum_a p_t(a)\tilde{\ell}_t(a) + \frac{\eta_t^2}{2}\sum_a p_t(a)\tilde{\ell}_t^2(a)\right) \qquad \left(1 + x \leq e^x\right)$$

Using the definition of $p'_{t+1}(a^*)$ in the l.h.s. of the inequality above we get

$$\exp\left(-\eta_t\tilde{\ell}_t(a^*)\right)\frac{p_t(a^*)}{p'_{t+1}(a^*)}$$

$$\leq \exp\left(-\eta_t\sum_a p_t(a)\tilde{\ell}_t(a) + \frac{\eta_t^2}{2}\sum_a p_t(a)\tilde{\ell}_t^2(a)\right)$$

Taking a logarithm and rearanging we get (23).

We now notice that

$$p_{t+1}(a^*)$$

$$= \frac{w_{t+1}(a^*)}{\sum_a w_{t+1}(a)}$$

$$= \frac{(1-\sigma)w_t(a^*)\exp\left(-\eta_t\tilde{\ell}_t(a^*)\right) + \frac{\sigma}{K}\sum_{a'} w_t(a')\exp\left(-\eta_t\tilde{\ell}_t(a')\right)}{\sum_a\left((1-\sigma)w_t(a)\exp\left(-\eta_t\tilde{\ell}_t(a)\right) + \frac{\sigma}{K}\sum_{a'} w_t(a')\exp\left(-\eta_t\tilde{\ell}_t(a')\right)\right)}$$

$$= \frac{(1-\sigma)w_t(a^*)\exp\left(-\eta_t\tilde{\ell}_t(a^*)\right) + \frac{\sigma}{K}\sum_{a'} w_t(a')\exp\left(-\eta_t\tilde{\ell}_t(a')\right)}{\sum_a w_t(a)\exp\left(-\eta_t\tilde{\ell}_t(a)\right)}$$

$$= \frac{(1-\sigma)p_t(a^*)\exp\left(-\eta_t\tilde{\ell}_t(a^*)\right) + \frac{\sigma}{K}\sum_{a'} p_t(a')\exp\left(-\eta_t\tilde{\ell}_t(a')\right)}{\sum_a p_t(a)\exp\left(-\eta_t\tilde{\ell}_t(a)\right)}$$

$$\geq (1-\sigma)p'_{t+1}(a^*) + 0$$

Bounding $p'_{t+1}(a^*)$ in (23) we get

$$\sum_a p_t(a)\tilde{\ell}_t(a) - \tilde{\ell}_t(a^*) \leq \frac{1}{\eta_t}\log\left(\frac{p_{t+1}(a^*)}{(1-\sigma)p_t(a^*)}\right) + \frac{\eta_t}{2}\sum_a p_t(a)\tilde{\ell}_t^2(a)$$

Let $I = [\tau_1, \tau_2]$; summing the above for all $i \in I$ we get

$$\sum_{t\in I}\sum_a p_t(a)\tilde{\ell}_t(a) - \sum_{t\in I}\tilde{\ell}_t(a^*)$$

$$\leq \sum_{t\in I}\frac{1}{\eta_t}\log\left(\frac{p_{t+1}(a^*)}{(1-\sigma)p_t(a^*)}\right) + \sum_{t\in I}\frac{\eta_t}{2}\sum_a p_t(a)\tilde{\ell}_t^2(a) \tag{24}$$

Now we focus on

$$\sum_{t \in I} \frac{1}{\eta_t} \log \left( \frac{p_{t+1}(a^*)}{(1-\sigma)p_t(a^*)} \right)$$

$$= \frac{\log p_{\tau_2+1}(a^*)}{\eta_{\tau_2}} - \frac{\log \left((1-\sigma)p_{\tau_1}(a^*)\right)}{\eta_{\tau_1}}$$

$$+ \sum_{t \in I \setminus \{\tau_1\}} \log \left( \frac{p_t^{1/\eta_{t-1}}(a^*)}{(1-\sigma)^{1/\eta_t} p_t^{1/\eta_t}(a^*)} \right)$$

$$\leq 0 - \frac{\log \left((1-\sigma)\frac{\sigma}{K}\right)}{\eta_{\tau_1}} + \sum_{t \in I \setminus \{\tau_1\}} \log \left( \frac{p_t^{\frac{1}{\eta_{t-1}} - \frac{1}{\eta_t}}(a^*)}{(1-\sigma)^{1/\eta_t}} \right)$$

$$\leq -\frac{\log \left((1-\sigma)\frac{\sigma}{K}\right)}{\eta_{\tau_1}} + \sum_{t \in I \setminus \{\tau_1\}} \left( \frac{1}{\eta_{t-1}} - \frac{1}{\eta_t} \right) \log \frac{\sigma}{K}$$

$$+ \sum_{t \in I \setminus \{\tau_1\}} \frac{1}{\eta_t} \log \frac{1}{1-\sigma}$$

$$= -U_{\tau_1} \frac{\log \left((1-\sigma)\frac{\sigma}{K}\right)}{\theta} + \frac{1}{\theta} \log \frac{\sigma}{K} \sum_{t \in I \setminus \{\tau_1\}} (U_{t-1} - U_t)$$

$$+ U_{\tau_2} \frac{|I| - 1}{\theta} \log \frac{1}{1-\sigma}$$

$$\leq -U_{\tau_2} \frac{\log \left((1-\sigma)\frac{\sigma}{K}\right)}{\theta} - \frac{U_{\tau_2}}{\theta} \log \frac{\sigma}{K} + U_{\tau_2} \frac{|I|-1}{\theta} \log \frac{1}{1-\sigma}$$

$$= U_{\tau_2} \left( \sqrt{TK} \log \frac{KT}{1 - \frac{1}{T}} + \sqrt{TK} \log(TK) + T^{3/2}\sqrt{K} \log \frac{1}{1 - \frac{1}{T}} \right)$$

$$= U_{\tau_2} \sqrt{TK} \left( \log \frac{KT^2}{T-1} + \log(TK) + \sqrt{T} \log \frac{T}{T-1} \right)$$

$$\leq U_{\tau_2} \sqrt{TK} \left( 2 \log \left( KT^2 \right) + 2 \frac{1}{\sqrt{T}} \right)$$

$$\leq U_{\tau_2} O\left( \sqrt{TK} \log(TK) \right) \tag{25}$$

where in the first inequality we use $p_t(a) \in \left[ \frac{\sigma}{K}, 1 \right]$. in the second inequality that $p_t(a) \geq \frac{\sigma}{K}$, in the equality after that the fact that $\eta_t = \frac{\theta}{U_t}$, in the second to last equality that $\sigma = \frac{1}{T}$ and $\theta = \frac{1}{\sqrt{KT}}$, and in the second to last inequality that $\log \frac{T}{T-1} \leq \frac{2}{T}$.

Now we bound

$$\sum_a p_t(a) \tilde{\ell}_t^2(a) = \sum_a p_t(a) \tilde{\ell}_t(a) \frac{\ell_t(a)}{p_t(a) + \xi} \leq U_t \sum_a \tilde{\ell}_t(a) \tag{26}$$

Using (25) and (26) in (24), and substituting $\eta_t = \theta/U_t$ we get

$$\sum_{t \in I} \sum_a p_t(a) \tilde{\ell}_t(a) - \sum_{t \in I} \tilde{\ell}_t(a^*) \leq U_{\tau_2} X + \frac{\theta}{2} \sum_{t \in I} \sum_a \tilde{\ell}_t(a) \tag{27}$$

Using a slight modification of [27, Corollary 1] we get that

$$\mathbb{P} \left[ \forall a \in [K] : \sum_{t \in I} \left( \tilde{\ell}_t(a) - \ell_t(a) \right) \leq U_{\tau_2} \frac{\log(K/\delta)}{2\xi} \right] \geq 1 - \delta \tag{28}$$

Now we bound

$$\sum_a p_t(a) \tilde{\ell}_t(a) = p_t(a_t) \frac{\ell_t(a_t)}{p_t(a_t) + \xi}$$

$$\geq \ell_t(a_t) - \ell_t(a_t) \frac{\xi}{p_t(a_t) + \xi} = \ell_t(a_t) - \xi \sum_a \tilde{\ell}_t(a) \tag{29}$$

We now combine all the above to bound the regret. Assume that (28) is true; for every $a^* \in [K]$:

$$\sum_{t \in I} \left( \ell_t(a_t) - \ell_t(a^*) \right)$$

$$\leq \sum_{t \in I} \sum_a p_t(a) \tilde{\ell}_t(a) + \xi \sum_{t \in I} \sum_a \tilde{\ell}_t(a) - \sum_{t \in I} \ell_t(a^*) \quad \text{(by (29))}$$

$$\leq \sum_{t \in I} \sum_a p_t(a) \tilde{\ell}_t(a) + \xi \sum_{t \in I} \sum_a \tilde{\ell}_t(a) - \sum_{t \in I} \tilde{\ell}_t(a^*)$$

$$+ U_{\tau_2} \frac{\log(K/\delta)}{2\xi} \quad \text{(by (28))}$$

$$\leq U_{\tau_2} O\left( \sqrt{TK} \log(TK) \right) + \frac{\theta}{2} \sum_{t \in I} \sum_a \tilde{\ell}_t(a)$$

$$+ \xi \sum_{t \in I} \sum_a \tilde{\ell}_t(a) + U_{\tau_2} \frac{\log(K/\delta)}{2\xi} \quad \text{(by (27))}$$

$$= U_{\tau_2} O\left( \sqrt{TK} \log(TK) \right) + U_{\tau_2} \frac{\log(K/\delta)}{2\xi}$$

$$+ \left( \frac{\theta}{2} + \xi \right) \sum_{t \in I} \sum_a \tilde{\ell}_t(a)$$

$$\leq U_{\tau_2} O\left( \sqrt{TK} \log(TK) \right) + U_{\tau_2} \frac{\log(K/\delta)}{2\xi}$$

$$+ \left( \frac{\theta}{2} + \xi \right) K \left( U_{\tau_2} |I| + U_{\tau_2} \frac{\log(K/\delta)}{2\xi} \right) \quad \text{(by (28))}$$

$$= U_{\tau_2} O\left( \sqrt{TK} \log(TK) \right) + U_{\tau_2} \sqrt{TK} \log(K/\delta)$$

$$+ \frac{1}{\sqrt{TK}} K \left( U_{\tau_2} T + U_{\tau_2} \sqrt{TK} \log(K/\delta) \right) \quad \left( \substack{\xi = \frac{1}{2\sqrt{TK}} \\ \theta = \frac{1}{\sqrt{TK}}} \right)$$

$$= U_{\tau_2} O\left( \sqrt{TK} \log(TK/\delta) \right)$$

$$+ U_{\tau_2} \frac{\sqrt{K}}{\sqrt{T}} \left( T + \sqrt{TK} \log(K/\delta) \right)$$

$$= U_{\tau_2} O\left( \sqrt{TK} \log(TK/\delta) + K \log(K/\delta) \right)$$

which proves the desired bound. □

We now proceed to prove Theorem F.3.

PROOF OF THEOREM F.3. Fix $N = \lceil L^{3/4} T^{1/4} \rceil$ and $K = \lceil T^{1/4}/L^{1/4} \rceil$. Let $\tilde{v}_i = i/N$ for $i \in [N]$ and $\tilde{b}_j = j/N$ for $j \in [N]$. For every $i \in [N]$ define $r_t^i : [K] \to \mathbb{R}_{\geq 0}$ such that

$$r_t^i(j) = \mathbb{1} \left[ \tilde{b}_{t,j} \geq d_t \right] \left( \chi_t \tilde{v}_i - \psi_t p(\tilde{b}_{t,j}, d_t) \right)$$

where $\tilde{b}_{t,j}$ is either $\tilde{b}_j$ or the safe bid of $\tilde{r}_t^i(\cdot)$.

Define for every round $i_t = \lceil v_t N \rceil$ (i.e. the $i$ that corresponds to the value that is closest and above to $v_t$). Fix $i \in [N]$ and let $\mathcal{T}_i \subseteq [T]$ be the rounds where $i = i_t$; let $T_i = |\mathcal{T}_i|$. Let $\mathcal{A}_i$ be an instance of Algorithm 4 which is run only on rounds $\mathcal{T}_i$. $\mathcal{A}_i$ has $K$ actions. The reward of the $j$-th action in round $t$ is $\tilde{r}_t^i(j)$. Let $j_t$ be the output of $\mathcal{A}_i$ in a round $t \in \mathcal{T}_i$, which we use to bid $\tilde{b}_{t,j_t}$. Because of Theorem F.4 we have that for every $\delta > 0$ with

probability at least $1 - \delta$, for every $\tau_1 \leq \tau_2$

$$\max_{j \in [K]} \sum_{t \in \mathcal{T}_i \cap [\tau_1, \tau_2]} \tilde{r}_t^i(j) - \sum_{t \in \mathcal{T}_i \cap [\tau_1, \tau_2]} \tilde{r}_t^i(j_t)$$

$$\leq U_{\tau_2} O\left(\left(\sqrt{T_i K} + K\right) \log \frac{T_i K}{\delta}\right)$$

We do the above process for every $i$ and use it as an algorithm[10]. Doing this for every $i$ and using the union bound we get that for every $\delta > 0$ with probability at least $1 - \delta$, for every $i \in [N]$ and $\tau_1 \leq \tau_2$

$$\max_{j \in [K]} \sum_{t \in \mathcal{T}_i \cap [\tau_1, \tau_2]} \tilde{r}_t^{i_t}(j) - \sum_{t \in \mathcal{T}_i \cap [\tau_1, \tau_2]} \tilde{r}_t^{i_t}(j_t)$$

$$\leq U_{\tau_2} O\left(\left(\sqrt{T_i K} + K\right) \log \frac{T_i K N}{\delta}\right)$$

$$= U_{\tau_2} O\left(\left(\sqrt{T_i \frac{T^{1/4}}{L^{1/4}}} + \frac{T^{1/4}}{L^{1/4}}\right) \log \frac{LT}{\delta}\right) \qquad \left(\begin{array}{l} N = \lfloor L^{3/4} T^{1/4} \rfloor \\ K = \lfloor T^{1/4} / L^{1/4} \rfloor \end{array}\right)$$

$$\tag{30}$$

Fix $\tau_1, \tau_2$. We want to use (30) to upper bound

$$\sup_{f \in \mathcal{F}} \sum_{t=\tau_1}^{\tau_2} r_t(f) - \sum_{t=\tau_1}^{\tau_2} r_t(\tilde{b}_{t,j_t})$$

Fix $f \in \mathcal{F}$ and define $j_1, \ldots, j_N \in [K]$ such that $\tilde{b}_{j_i}$ (recall $\tilde{b}_j = j/K$) is the bid that is above $f(v)$ for every $v \in (v_{i-1}, v_i]$ and as small as possible, i.e., for all $i \in [N]$ it holds

$$\tilde{b}_{j_i} = \left\lceil K \sup_{v \in (\tilde{v}_{i-1}, \tilde{v}_i]} f(v) \right\rceil / K$$

We notice that for all $i \in [N]$ and $v \in (v_{i-1}, v_i]$: $\tilde{b}_{j_i} \in [f(v), f(v) + \frac{L}{N} + \frac{1}{K}]$ (by $L$-Lipschitz of $f$). We now have

$$r_t(f) = \mathbb{1}\left[f(v_t) \geq d_t\right]\left(\chi_t v_t - \psi_t p\left(f(v_t), d_t\right)\right)$$

$$\leq \mathbb{1}\left[\tilde{b}_{j_{i_t}} \geq d_t\right]\left(\chi_t \tilde{v}_{i_t} - \psi_t p\left(\tilde{b}_{j_{i_t}} - \frac{L}{N} - \frac{1}{K}, d_t\right)\right)^+$$

$$\leq \mathbb{1}\left[\tilde{b}_{j_{i_t}} \geq d_t\right]\left(\chi_t \tilde{v}_{i_t} - \psi_t p\left(\tilde{b}_{j_{i_t}}, d_t\right)\right)^+ + \psi_t\left(\frac{L}{N} + \frac{1}{K}\right)$$

$$\leq \tilde{r}_t^{i_t}(j_{i_t}) + \psi_t\left(\frac{L}{N} + \frac{1}{K}\right)$$

where the last inequality follows because the bid that corresponds to $j_{i_t}$ in round $t$ is the safe bid of that round whose reward is non-negative and at least as good as $\tilde{b}_{j_{i_t}}$'s. Summing the above over $t \in [\tau_1, \tau_2]$ and taking a sup over $f$ and a max over $j_1, \ldots, j_N$ we

---

[10]For rounds where $v_t = 0$ we have not defined an algorithm; in those rounds we can bid 0 (which is optimal) to guarantee no regret; for simplicity however we assume that $v_t > 0$.

get

$$\sup_{f \in \mathcal{F}} \sum_{t=\tau_1}^{\tau_2} r_t(f)$$

$$\leq \max_{j_1, \ldots, j_N} \sum_{t=\tau_1}^{\tau_2} \tilde{r}_t^{i_t}(j_{i_t}) + U_{\tau_2} T\left(\frac{L}{N} + \frac{1}{K}\right)$$

$$= \sum_{i \in [N]} \max_{j \in [K]} \sum_{t \in \mathcal{T}_i \cap [\tau_1, \tau_2]} \tilde{r}_t^{i_t}(j) + U_{\tau_2} T\left(\frac{L}{N} + \frac{1}{K}\right)$$

$$\leq \sum_{i \in [N]} \max_{j \in [K]} \sum_{t \in \mathcal{T}_i \cap [\tau_1, \tau_2]} \tilde{r}_t^{i_t}(j) + U_{\tau_2} 2 T^{\frac{3}{4}} L^{\frac{1}{4}} \qquad \left(\begin{array}{l} N \geq L^{\frac{3}{4}} T^{\frac{1}{4}} \\ K \geq T^{\frac{1}{4}} / L^{\frac{1}{4}} \end{array}\right)$$

$$\leq \sum_{t \in [\tau_1, \tau_2]} \tilde{r}_t^{i_t}(j_t)$$

$$+ \sum_{i \in [N]} U_{\tau_2} O\left(\left(\sqrt{T_i \frac{T^{\frac{1}{4}}}{L^{\frac{1}{4}}}} + \frac{T^{\frac{1}{4}}}{L^{\frac{1}{4}}}\right) \log \frac{LT}{\delta}\right)$$

$$+ U_{\tau_2} 2 T^{\frac{3}{4}} L^{\frac{1}{4}} \qquad \text{(by (30))}$$

$$= \sum_{t \in [\tau_1, \tau_2]} \tilde{r}_t^{i_t}(j_t)$$

$$+ U_{\tau_2} O\left(\left(L^{\frac{1}{4}} T^{\frac{3}{4}} + (TL)^{\frac{1}{2}}\right) \log \frac{LT}{\delta} + T^{\frac{3}{4}} L^{\frac{1}{4}}\right)$$

$$= \sum_{t \in [\tau_1, \tau_2]} \tilde{r}_t^{i_t}(j_t)$$

$$+ U_{\tau_2} O\left(\left(L^{\frac{1}{4}} T^{\frac{3}{4}} + (TL)^{\frac{1}{2}}\right) \log \frac{LT}{\delta}\right) \tag{31}$$

where the second to last inequality we substituted $N = \lfloor L^{3/4} T^{1/4} \rfloor$ and we used the fact that $\sum_i \sqrt{T_i} \leq \sqrt{NT}$ since $\sum_i T_i = T$. By noticing that $\tilde{r}_t^i(j_t) \leq r_t(\tilde{b}_{t,j_t}) + U_t(1/K + 1/N)$ and bounding $T(1/N + 1/K) = O(L^{1/4} T^{3/4})$ we get the high probability interval regret bound on the rewards $r_t(\cdot)$. Using Theorem 2.1 we get the desired result.

The tight satisfaction of the ROI constraint follows by using Lemma 4.2. The first algorithm is the one we describe above. The second algorithm is the primal algorithm described above with $\chi_t = \psi_t = 1$, which achieves $Q(\tau, \delta) = \tau \beta - O\left((L^{1/4} \tau^{3/4} + \tau^{1/2} L^{1/2}) \log \frac{LT}{\delta}\right)$. $\square$

# G  Polynomial time algorithm for full information feedback

In this section, we present a polynomial time algorithm that has regret guarantees matching the ones in Theorems 3.3 and 4.1, when the value and the highest competing bid are independent.

THEOREM G.1. *Assume that the value $v_t$ and highest competing bid $d_t$ are sampled independently. There is a polynomial time algorithm with full information feedback for value or quasi-linear utility maximization when the payment function satisfies Assumption 3.1 such that, for every $\delta > 0$ with probability at least $1 - \delta$, the algorithm has regret against the class of $L$-Lipschitz continuous functions and ROI violation at most $\frac{1}{\beta \rho} O(\sqrt{T \log(TL/\delta)})$. In addition, if $\beta = \Omega\left(T^{-1/2+\varepsilon} \log(TL/\delta)\right)$ for some constant $\varepsilon > 0$, it can be turned*

*into an algorithm with exact ROI satisfaction and with regret that is $1/\beta$ times worse.*

The algorithm for the above theorem is similar to the one in Theorem F.3: discretize the values and run a separate online learning algorithm for each of those values. The important difference is that, instead of running/updating each algorithm only when we observe its corresponding value, we run every algorithm in every round $t$, even if the value $v_t$ is completely different than the one the algorithm represents. This allows each algorithm to run for many more rounds, making it 'learn' faster. The assumption that $v_t$ and $d_t$ are independent is crucial since the $d_t$ of every round can be used for every algorithm.

We note that the above technique cannot directly be used for bandit information, since we do not observe $d_t$ and therefore the observed reward is dependent on the bidding. [5] use this technique, along with the assumption that the value distribution is known, to get regret bounds that would imply to $\tilde{O}\left(T^{2/3}\right)$ regret for bandit information. However, their bounds are only in expectation and for the entire horizon, not every interval. We leave as future work extending these techniques to get regret matching the one in Theorem 5.1.

PROOF. We use a scheme similar to the one used in the proof of Theorem F.3. Fix $N = \lfloor LT \rfloor$ and $K = T$. Let $\tilde{v}_i = i/N$ for $i \in [N]$ and $\tilde{b}_j = j/N$ for $j \in [N]$. For every $i \in [N]$ define $r_t^i : [K] \to \mathbb{R}_{\geq 0}$ such that

$$r_t^i(j) = \mathbb{1}\left[\tilde{b}_{t,j} \geq d_t\right]\left(\chi_t \tilde{v}_i - \psi_t p(\tilde{b}_{t,j}, d_t)\right)$$

where $\tilde{b}_{t,j}$ is either $\tilde{b}_j$ or the safe bid (Assumption 3.1) of $\tilde{r}_t^i(\cdot)$.

Define for every round $i_t = \lceil v_t N \rceil$ (i.e. the $i$ that corresponds to the value that is closest and above to $v_t$). For every $i \in [N]$ let $\mathcal{A}_i$ be an instance of the algorithm in Theorem 3.5 for $\Delta = 1$ with $K$ actions that has also been modified to have low interval regret, using Theorem 3.7. Unlike the bandit setting, here we run $\mathcal{A}_i$ even in rounds where $i_t \neq i$. In every round $\mathcal{A}_i$ outputs an action $j_t^i \in [K]$ which corresponds to the bid $\tilde{b}_{t,j_t^i}$. The reward of the $j$-th action in round $t$ is $\tilde{r}_t^i(j)$. The bid we use in round $t$ is the one suggested by algorithm $\mathcal{A}_{i_t}$, $\tilde{b}_{t,j_t^{i_t}}$.

Using Theorems 3.5 and 3.7, an application of Azuma's inequality and a union bound over $i$, we have that with probability at least $1 - \delta$ for every $1 \leq \tau_1 < \tau_2 \leq T$ and every $i \in [N]$:

$$\max_{j \in [K]} \sum_{t=\tau_1}^{\tau_2} \tilde{r}_t^i(j) - \sum_{t=\tau_1}^{\tau_2} \tilde{r}_t^i(j_t^i) \leq U_{\tau_2} O\left(\sqrt{T \log(TKN/\delta)}\right) \quad (32)$$

Fix $\tau_1, \tau_2$. To get our regret bound we have to upper bound

$$\sup_{f \in \mathcal{F}} \sum_{t=\tau_1}^{\tau_2} r_t(f) - \sum_{t=\tau_1}^{\tau_2} \sum_i \mathbb{1}\left[i_t = i\right] r_t(\tilde{b}_{t,j_t^i})$$

Fix $f \in \mathcal{F}$ and define $j_1, \ldots, j_N \in [K]$ such that for all $i \in [N]$

$$\tilde{b}_{j_i} = \left\lceil K \sup_{v \in (\tilde{v}_{i-1}, \tilde{v}_i]} f(v) \right\rceil / K$$

We notice that for all $i \in [N]$ and $v \in (v_{i-1}, v_i]$: $\tilde{b}_{j_i} \in [f(v), f(v) + \frac{L}{N} + \frac{1}{K}]$ (by L-Lipschitz of $f$). We now have

$$r_t(f) = \mathbb{1}\left[f(v_t) \geq d_t\right]\left(\chi_t v_t - \psi_t p(f(v_t), d_t)\right)$$

$$\leq \mathbb{1}\left[\tilde{b}_{j_{i_t}} \geq d_t\right]\left(\chi_t \tilde{v}_{i_t} - \psi_t p\left(\tilde{b}_{j_{i_t}} - \frac{L}{N} - \frac{1}{K}, d_t\right)\right)^+$$

$$\leq \mathbb{1}\left[\tilde{b}_{j_{i_t}} \geq d_t\right]\left(\chi_t \tilde{v}_{i_t} - \psi_t p\left(\tilde{b}_{j_{i_t}}, d_t\right)\right)^+ + \psi_t\left(\frac{L}{N} + \frac{1}{K}\right)$$

$$\leq \tilde{r}_t^{i_t}(j_{i_t}) + \psi_t\left(\frac{L}{N} + \frac{1}{K}\right)$$

where in the second to last inequality we used the fact that $p(\cdot, d)$ is 1-Lipschitz and in the last inequality follows because the bid that corresponds to $j_{i_t}$ in round $t$ is the safe bid of that round whose reward is non-negative and at least as good as $\tilde{b}_{j_{i_t}}$'s. Summing the above over $t \in [\tau_1, \tau_2]$ and taking a sup over $f$ and a max over $j_1, \ldots, j_N$ we get

$$\sup_{f \in \mathcal{F}} \sum_{t=\tau_1}^{\tau_2} r_t(f) \leq \max_{j_1, \ldots, j_N} \sum_{t=\tau_1}^{\tau_2} \tilde{r}_t^{i_t}(j_{i_t}) + U_{\tau_2} T\left(\frac{L}{N} + \frac{1}{K}\right)$$

$$\leq \sum_{i \in [N]} \max_{j^* \in [K]} \sum_{t=\tau_1}^{\tau_2} \mathbb{1}\left[i_t = i\right] \tilde{r}_t^i(j^*) + 2U_{\tau_2} \quad (33)$$

where in the last inequality we used that $N \geq \lfloor TL \rfloor$ and $K = T$.

We now prove a high probability bound on $\sum_{t=\tau_1}^{\tau_2} \mathbb{1}\left[i_t = i\right] \tilde{r}_t^i(j^*)$, for a fixed $j^*$. Using Azuma's inequality we can prove that for all $\delta > 0$ with probability at least $1 - \delta$,

$$\sum_{t=\tau_1}^{\tau_2} \mathbb{1}\left[i_t = i\right] \tilde{r}_t^i(j^*) \leq \sum_{t=\tau_1}^{\tau_2} \mathbb{P}\left[i_t = i\right] \tilde{r}_t^i(j^*) + O\left(U_{\tau_2}\sqrt{T \log(1/\delta)}\right)$$

Note that to use Azuma's inequality we have to rely on the fact that $i_t$ and $\tilde{r}_t^i(j^*)$ are independent conditioned on the history of rounds up to $t - 1$, since $\tilde{r}_t^i(j^*)$ does not depend on $v_t$ but only $d_t$.

By letting $\mathbb{P}\left[i_t = i\right] = q_i$ and taking a union bound over all $j* \in [K]$ and all $i \in [N]$, we have that with probability at least $1 - \delta$ for all $j_1^*$, $i$, and $\tau_1, \tau_2$ we have

$$\sum_{t=\tau_1}^{\tau_2} \mathbb{1}\left[i_t = i\right] \tilde{r}_t^i(j^*) \leq \sum_{t=\tau_1}^{\tau_2} q_i \tilde{r}_t^i(j^*) + O\left(U_{\tau_2}\sqrt{T \log(TKN/\delta)}\right)$$

Substituting $N$ and $K$, the above makes (33)

$$\sup_{f \in \mathcal{F}} \sum_{t=\tau_1}^{\tau_2} r_t(f) \leq \sum_{i \in [N]} q_i \max_{j^* \in [K]} \sum_{t=\tau_1}^{\tau_2} \tilde{r}_t^i(j^*) + O\left(U_{\tau_2}\sqrt{T \log(TL/\delta)}\right)$$

$$(34)$$

We now bound $q_i \max_{j^*} \sum_{t=\tau_1}^{\tau_2} \tilde{r}_t^i(j^*)$ for some $i$. By (32) we have

$$q_i \max_{j^*} \sum_{t=\tau_1}^{\tau_2} \tilde{r}_t^i(j^*) \leq q_i \sum_{t=\tau_1}^{\tau_2} \tilde{r}_t^i(j_t^i) + q_i U_{\tau_2} O\left(\sqrt{T \log(TL/\delta)}\right)$$

$$\leq \sum_{t=\tau_1}^{\tau_2} \mathbb{1}\left[i_t = i\right] \tilde{r}_t^i(j_t^i) + q_i U_{\tau_2} O\left(\sqrt{T \log(TKN/\delta)}\right)$$

$$+ q_i U_{\tau_2} O\left(\sqrt{T \log(TL/\delta)}\right)$$

$$= \sum_{t=\tau_1}^{\tau_2} \mathbb{1}\left[i_t = i\right] \tilde{r}_t^i(j_t^i) + q_i U_{\tau_2} O\left(\sqrt{T \log(TL/\delta)}\right)$$

where in the last inequality we used Azuma's inequality, which heavily depends on the fact that $i_t$ and $\tilde{r}_t^i(j_t^i)$ are independent

conditioned on the history of rounds up to $t-1$, since $\tilde{r}^i_t(j^i_t)$ does not depend on $v_t$ but only $d_t$. Plugging the above into (34) we get

$$\sup_{f \in \mathcal{F}} \sum_{t=\tau_1}^{\tau_2} r_t(f) \leq \sum_{i \in [N]} \sum_{t=\tau_1}^{\tau_2} \mathbb{1}\,[i_t = i]\,\tilde{r}^i_t(j^i_t)$$
$$+ U_{\tau_2} O\left(\sqrt{T \log(TL/\delta)}\right)$$

By noticing that $\tilde{r}^i_t(j_t) \leq r_t(\tilde{b}_{t,j_t}) + U_t(1/K + 1/N)$ and bounding $T(1/N + 1/K) = O(1)$ we get the $U_{\tau_2} O\left(\sqrt{T \log(TL/\delta)}\right)$ high probability interval regret bound on the rewards $r_t(\cdot)$. Using Theorem 2.1 we get the desired result.

The tight satisfaction of the ROI constraint follows by using Lemma 4.2. The first algorithm is the one we describe above. The second algorithm is the primal algorithm described above with $\chi_t = \psi_t = 1$, which achieves $Q(\tau, \delta) = \tau\beta - O(\sqrt{\tau \log(\tau L/\delta)})$. □

