# OpenReview forum: "No-Regret Algorithms in non-Truthful Auctions with Budget and ROI Constraints"
_ACM.org/TheWebConf/2025/Conference — WWW 2025 Oral_

### Official Review · Reviewer_si63 · 2024-11-30

**Novelty:** 5
**Technical Quality:** 5

**Review:**

Summary:
This paper develops a bidding algorithm that achieves root T regret when compared to the best Lipschitz bidding function. This is a stronger baseline than has been used in past work. They solve for this algorithm by using a  primal/dual framework and finding a new primal algorithm which gives good interval regret, a strong notion of regret.

They end with two smaller results. The first shows how to turn a no regret algorithm which satisfies budget and RoI in expectation into one that satisfies it with high probability while incurring a dependency on the budget and RoI parameters. Finally, they show that in the case of bandit feedback, even if the bidder only cares about winning the item and satisfying the constraints, they cannot achieve similar regret bounds.

Strengths:

- The authors do a good job at expressing the parts of their analysis that are novel and of independent interest
- The paper is well written and easy to follow
- The problem of designing automated bidding is of obvious interest to the community.

Weaknesses:

- Without knowing the literature it isnt obvious whether the addition of Lipschitz baseline instead of a constant multiple is a large contribution

**Questions:**

Questions:

1. The definition of OPT does not (as far as i can tell) allow the algorithm to depend on how much of its budget has been depleted so far. Does the budget constraint need to hold in expectation or ex-post for the baseline?

2. Does the agent know the (unconditioned of d_t) distribution of their own values or are they learning this as well? Can the agent do anything useful in terms of regret with this information if they did have it?

**Reviewer Confidence:**

3: The reviewer is confident but not certain that the evaluation is correct

**Scope:**

4: The work is relevant to the Web and to the track, and is of broad interest to the community

---

### Official Review · Reviewer_ufD7 · 2024-12-01

**Novelty:** 6
**Technical Quality:** 6

**Review:**

**Summary**:

This paper addresses a general and flexible model for autobidding with budget and Return-on-Investment (ROI) constraints. The authors adopted a primal/dual approach and derived regret bounds across both full and bandit feedback settings (depending on whether the highest bid will be revealed at each round of the game). Overall, I believe the paper is technically sound and delivers significant theoretical insights and rigorously tackles complex technical challenges. However, the paper is not well presented and definitely dense/challenging for non-expert readers.

**Strengths**:

1. Generality of the Model:

- The model encompasses a wide range of auction settings and covers multiple buyer objectives (e.g., value-maximizing and quasi-linear utility) and constraints (e.g., budget and ROI), making the work broadly applicable.

2. Solid Theoretical Results:
- The regret bounds are near-optimal for full-information settings ( $\tilde{O}(\sqrt{T}) $).
- The lower bound for bandit settings ( $\Omega(T^{2/3})$ ) and the complementary upper bound ( $\tilde{O}(T^{3/4})$ ) are both novel and insightful.

3. Non-Trivial Primal/Dual Proof Techniques:
- The execution of the primal/dual framework is technically sophisticated, particularly in designing the primal algorithm.
- The use of Lipschitz bidding functions as a benchmark adds depth to the analysis and extends beyond traditional constant pacing multipliers.

**Weakness**:

- The paper is heavily focused on technical challenges and methodologies, with minimal emphasis on practical implications or broader insights.

- Some sections, such as Sections 2 and 3, are overly long and difficult to parse. Additionally, the transitions from the problem formulation to the algorithmic solutions in Section 2 are abrupt and fail to clearly guide the reader through the logical progression of the paper.

**Questions:**

- The problem setting: if we know some partial information about $ v_t $ or $ d_t $ (e.g., $v_t$'s distribution is unknown but $ d_t $'s is known), is it possible to strength your results? More generally, if we carefully model the correlation between $ v_t $ and $ d_t $, any potential improvement of the regret bounds (in both full and bandit feedback settings)?
- Consider to defer some technical discussions in Section 2 to later parts or appendix (e.g., the discussions of the primal/dual framework in Section 2)

**Reviewer Confidence:**

2: The reviewer is willing to defend the evaluation, but it is likely that the reviewer did not understand parts of the paper

**Scope:**

4: The work is relevant to the Web and to the track, and is of broad interest to the community

---

### Official Review · Reviewer_MdWu · 2024-12-02

**Novelty:** 5
**Technical Quality:** 5

**Review:**

This paper considers the problem of online auto-bidding in non-truthful auctions, focusing on optimizing ad campaigns subject to budget and ROI constraints. They propose a new algorithm for the full-information setting, achieving near-optimal regret concerning the best Lipschitz bidding function. Here the full-information setting refers to the setting where the highest competing bid is revealed after each round. This algorithm proves effective for various auction types, including first price, second price or their combinations, and bidder utility functions. A lower bound of Ω(T^(2/3)) is established in the bandit setting. This shows a disparity between full-information and bandit settings. The paper addresses several technical challenges, including Lagrangian maximization, discretization error, time-varying ranges, and interval regret.

Strengths: Compared to the previous literature, the biggest advancement is that this work uses the best Lipschitz bidding function as the benchmark, and considers a general class of auctions. This poses several technical challenges, including discretization and dealing with time-varying ranges. This paper provides several non-trivial techniques to address those challenges.

Weaknesses: Most algorithmic framework and analysis ideas are based on previous work. The prime-dual framework is based on [10]. The discretization of the family of Lipschitz bidding functions is based on [18].

In the introduction (line 77-85), the paper highlights that previous work either focuses on truthful auctions or non-truthful auctions with a benchmark much weaker than the best Lipschitz bidding function. The technical challenge also highlights the difficulty of finding a good finite cover of the family of Lipschitz bidding functions.

However, [18] also considers a special non-truthful auction, the first price auction, with a benchmark as the best Lipschitz bidding function. The paper only mentions [18] when it introduces the discretization techniques of the family of Lipschitz bidding functions. It seems that the main technical novelty in this paper is the transformation of bids to "safe" bids, which is crucial for analyzing general non-truthful auctions with general price functions.

To provide a clearer context for the paper's contributions, it would be beneficial to discuss relevant prior work, including [18], earlier in the introduction and offer a more direct comparison. Furthermore, the proposed algorithms are complex. While the paper provides theoretical insights, it would be better to discuss the practical implementation challenge of these algorithms.

Minor comments: in many places, this paper mentions the results can be easily extended to other settings. It would be better to explicitly add the extension and the proof in the paper.

**Questions:**

Could the author offer a more direct comparison to [18] regarding the technique?

**Reviewer Confidence:**

3: The reviewer is confident but not certain that the evaluation is correct

**Scope:**

4: The work is relevant to the Web and to the track, and is of broad interest to the community

---

### Official Review · Reviewer_DJYh · 2024-12-03

**Novelty:** 5
**Technical Quality:** 6

**Review:**

This paper considers the problem of learning an optimal bid function online in a first-price or hybrid first- and second-price auction. The authors assume the bidder's value is drawn iid each round, as is the highest competing bid from other agents. In addition to maximizing utility (or value) over time, the learner is subject to an interim budget constraint and an ROI constraint on the ratio of value to payments. The authors consider several variants of the problem, including both utility and value maximizing bidders, first-price or hybrid auctions, and both bandit and full feedback.

The main result is an online learning result that has low regret with respect to the best Lipschitz bidding function, which improves over existing results which consider a much more limited family of bidding functions as the benchmark. The main techniques are a primal-dual framework and a tree-based algorithm for effectively covering the space of bidding functions, and a couple other tweaks to adapt these ideas to the specific setting of onilne auction bids. They additionally show that the full-information regret bounds are tight, while offering looser bounds for the bandits setting.

Strengths:
 • Offers a clear technical improvement over existing work, which considered a weaker benchmark.
 • Gives tight bounds for the bandits setting.
 • Several of the results along the way might be of independent interest. In particular, the reduction from standard regret to interval regret and that from approximate to exact ROI constraint satisfaction.

Main weaknesses:
• Despite the authors' efforts, the sheer amount of technical material (and the relatively complex algorithmic approach) renders the paper pretty difficult to follow. Much of the technical discussion is just not really possible to understand without digging into the math in the appendices.
• Another specific example was Theorem 3.2. It seems to require knowledge of a "safe" bid, but it is not clear how that knowledge is attained unless the dual variables are revealed before bidding. But elsewhere the dual variables are discussed as "adversarial", seemingly implying that they are not known to the bidder in advance. Which is it?

**Questions:**

I'd appreciate some clarification on the second weakness mentioned in the main review (re: Thm. 3.2).

**Reviewer Confidence:**

2: The reviewer is willing to defend the evaluation, but it is likely that the reviewer did not understand parts of the paper

**Scope:**

4: The work is relevant to the Web and to the track, and is of broad interest to the community

---

### Official Review · Reviewer_gsar · 2024-12-03

**Novelty:** 6
**Technical Quality:** 6

**Review:**

This paper studies the online learning problem of bidding in a repeated first- or second-price auction where the bidder's value and the highest rivals' bid are drawn i.i.d. from an unknown distribution in each round, and the bidder needs to maximize either the quasi-linear utility or the winning values, subject to a total budget constraint and an ROI constraint.  The benchmark is the retrospectively optimal L-Lipschitz bidding function, and the algorithm given here achieves an \tilde O(\sqrt T) regret in the full information setting.  The benchmark is much richer than the class of pacing multipliers, and the regret bound improves over the \tilde O(T^{2/3}) bound obtainable from standard arguments.  For the bandit setting, the paper gives an \Omega(T^{2/3}) lower bound, and gives an algorithm achieving \tilde O(T^{3/4}) regret.

The algorithm builds on many recent developments.  The authors clearly indicate their own contributions, which include:
1. a learning algorithm that adapts to reward functions with dynamically changing upper and lower bounds, with improved regret over the existing approach;
2. a way of handling the discrete approximation of Lipschitz functions that avoids the problem of drastically different utilities with slightly different values in auctions --- the solution is also somewhat specific to auctions, where it is observed that a ``safe bid'' exists and can be used to replace bids that may cause such jumps;
3. a reduction from interval regret minimization to ordinary regret minimization, by way of a meta-algorithm.

The lower bound is also nontrivial, building on an argument from Kleinberg & Leighton.

Even though the framework is mostly from existing works, these contributions are nontrivial, and contributions 1 & 3 are particularly likely to find other applications.

The paper is masterfully written.  It is a delightful reading, which is unusual for a work that builds on so much existing machinery.



Minor comments:

Section 3.1, first paragraph: "takes into advantage" should be "takes advantage of"

Lemma 4.2: I believe in the last paragraph, the condition under which algorithm A_1 should be run has some typos.

Section 5: paragraph after Theorem 5.1, last sentence seems to refer to the ratio instead of the "value".

**Questions:**

1. The values are assumed to be from [0, 1]; usually this assumes some normalization, and the regret, which is additive, only makes sense in this scale.  Given this, is it still justified to assume \gamma = 1, by assuming one may rescale the values?

2. I got lost amid all these algorithms why in Section 3.1 the algorithm can know \chi_t and \psi_t before bidding in round t.  Could you explain?

**Reviewer Confidence:**

3: The reviewer is confident but not certain that the evaluation is correct

**Scope:**

4: The work is relevant to the Web and to the track, and is of broad interest to the community